# Mitochondrial membrane proteins and VPS35 orchestrate selective removal of mtDNA

Ayesha Sen[1], Sebastian Kallabis[2], Felix Gaedke[2], Christian Jüngst [2], Julia Boix[1,3], Julian Nüchel [4], Kanjanamas Maliphol[1], Julia Hofmann[1], Astrid C. Schauss[2], Marcus Krüger[2], Rudolf J. Wiesner [1,2,3] & David Pla-Martín [1,3] ✉

Understanding the mechanisms governing selective turnover of mutation-bearing mtDNA is fundamental to design therapeutic strategies against mtDNA diseases. Here, we show that specific mtDNA damage leads to an exacerbated mtDNA turnover, independent of canonical macroautophagy, but relying on lysosomal function and ATG5. Using proximity labeling and Twinkle as a nucleoid marker, we demonstrate that mtDNA damage induces membrane remodeling and endosomal recruitment in close proximity to mitochondrial nucleoid sub-compartments. Targeting of mitochondrial nucleoids is controlled by the ATAD3-SAMM50 axis, which is disrupted upon mtDNA damage. SAMM50 acts as a gatekeeper, influencing BAK clustering, controlling nucleoid release and facilitating transfer to endosomes. Here, VPS35 mediates maturation of early endosomes to late autophagy vesicles where degradation occurs. In addition, using a mouse model where mtDNA alterations cause impairment of muscle regeneration, we show that stimulation of lysosomal activity by rapamycin, selectively removes mtDNA deletions without affecting mtDNA copy number, ameliorating mitochondrial dysfunction. Taken together, our data demonstrates that upon mtDNA damage, mitochondrial nucleoids are eliminated outside the mitochondrial network through an endosomal-mitophagy pathway. With these results, we unveil the molecular players of a complex mechanism with multiple potential benefits to understand mtDNA related diseases, inherited, acquired or due to normal ageing.

The accumulation over time of mutations in the mitochondrial genome (mtDNA) is a common process that has been shown to occur in many tissues as one of the hallmarks of ageing[1]. mtDNA is present in thousands of copies per cell, hence, impairment of mitochondrial function is observed only when the percentage of mutated mtDNA molecules surpasses a specific threshold[2]. Cells possess a plethora of quality control mechanisms to survey the intactness of DNA, RNA, and proteins, but also of entire organelles. In addition to bulk autophagy, which is responsible for the continuous and non-selective turnover of cellular material, specific mechanisms to remove malfunctioning organelles upon damage are known. The process of mitophagy has been investigated extensively in recent years as an important salvage

[1]Center for Physiology and Pathophysiology, Faculty of Medicine and University Hospital Cologne, University of Cologne, Cologne, Germany. [2]Cologne Excellence Cluster on Cellular Stress Responses in Ageing-associated Diseases (CECAD), University of Cologne, Cologne, Germany. [3]Center for Molecular Medicine Cologne, University of Cologne, Cologne, Germany. [4]Center for Biochemistry, Faculty of Medicine and University Hospital Cologne, University of Cologne, Cologne, Germany. ✉e-mail: dplamart@uni-koeln.de

pathway to remove dysfunctional mitochondria[3]. Upon mitochondrial damage, PINK1 is presented at the mitochondrial membrane signaling the part of the network prompted to degradation. PINK1 stabilization induces recruitment of the cytosolic protein Parkin together with other autophagy proteins, and initiates the formation of an autophagosome which ends with the engulfment of damaged mitochondria[3]. Mitochondrial dynamics are important processes surveying mitochondrial quality. Mitochondrial fission facilitates removal of the mitochondrial parts with impaired function, while mitochondrial fusion is required for mtDNA replication. Thus, knockout of key players of mitochondrial fusion induce mtDNA instability by causing a rapid accumulation of mtDNA alterations over time[4].

In addition to canonical forms of mitophagy, other special pathways have been described. A process with a high level of specificity involving mitochondrial-derived vesicles (MDVs) was shown to remove not the complete organelle, but rather mitochondrial fragments containing specific cargo[5]. This mechanism requires the coordination of mitochondrial dynamics, mitophagy, and also the vacuolar protein sorting (VPS) or retromer complex. Thus, changes in the mitochondrial membrane potential and the oxidation state of mitochondrial subcompartments induce the curvature of the membrane, followed by recruitment of PINK1 and Parkin[6]. The retromer, formed by the proteins VPS26, VPS29 and VPS35, provides the force to generate a vesicle, which is delivered to lysosomes or peroxisomes, in a process independent of the autophagy proteins ATG5 or LC3[7–9]. Other noncanonical pathways include direct transfer of mitochondrial parts to endosomes[10]. In these cases, mitochondrial damage triggers recruitment of Parkin, which is recognized by RAB5 endosomes engulfing mitochondrial parts guided towards the lysosomal compartment, where degradation occurs. Despite recent advances, little is known about how mitochondria contact the endo-lysosome system. In yeast, endo-lysosome-mitochondria tethering is regulated by physical interaction between the mitochondrial protein Tom40 and the endosomal proteins Vps39 and Vps13[11]. In mammalian cells, MFN2 and GDAP1 have been shown to mediate endo-lysosome-mitochondria contacts through RAB7 and LAMP1, respectively[12,13].

Many inherited forms of neurodegenerative diseases are examples for insufficient mitochondrial quality control. Several forms of Charcot-Marie-Tooth neuropathy have been linked to mutations in genes encoding for GDAP1, MFN2, and RAB7[14]. However, whether mitophagy impairment is the primary cause of the disease or a secondary effect is still not clear. The strongest link between neurodegeneration and mitochondrial quality control defects relies on Parkinson's disease (PD). PD is caused by the specific degeneration of dopaminergic neurons, which have been comprehensively shown to be a hotspot for the accumulation of large-scale mtDNA deletions during normal ageing[15,16]. Thus, not only mutations of specific mitophagy receptors like PINK1 and Parkin, but also malfunction of the lysosomal proteins ATP13A2 and LAMP3, along with mutations in the retromer component VPS35[17] cause familial forms of PD. Comparably, mutations in essential genes for mtDNA replication and maintenance, such as Twinkle and POLγ, lead to several forms of severe diseases including Parkinsonism, Progressive External Ophthalmoplegia, and Spinocerebellar Ataxia[18]. Therefore, therapeutic approaches to increase mitochondrial quality control and counteract the progression of mitochondrial-related diseases have been attempted[19,20]. Nevertheless, understanding the molecular mechanisms guiding the specificity of the different forms for the mitochondrial quality control systems is essential to design precise strategies and avoiding the activation of undesirable effects.

Expressing the mitochondrial helicase Twinkle, bearing disease-related dominant negative mutations, have been used to induce mtDNA instability and study mtDNA-related diseases[21]. In mouse models, expression of the Twinkle mutation p.K320E (from now on K320E), accelerates the accumulation of mtDNA deletions in postmitotic tissues[22–24] and induces mtDNA depletion in proliferating cells[25,26]. Using a combination of genetic and chemical tools, we have now identified the proteins involved in a mechanism for the targeting and degradation of mutated mtDNA through mitochondria-endosome transfer without affecting the mitochondrial pool. Expression of K320E or chemical damage triggers the elimination of altered mtDNA molecules through endosomes, a process controlled by the interaction between mitochondrial nucleoids with the inner membrane protein ATAD3 and the outer membrane and translocase protein SAMM50. On the mitochondrial surface, SAMM50 regulates the distribution of BAK foci and balances nucleoid release, which are later captured by VPS35 endosomes. SAMM50 and VPS35 are essential to provide the required selectivity and specificity for mtDNA elimination. While VPS35 is necessary to avoid the activation of an exacerbated mitophagy response, SAMM50 acts as a sentinel, surveillant proper mtDNA transfer to VPS35 endosomes. Finally, we demonstrate that stimulation of lysosomal function by rapamycin in vivo is sufficient to specifically remove deleted mtDNA, without affecting the total mtDNA copy number, strengthening modulation of autophagy as an approach to counteract the accumulation of mutations in mtDNA, as observed in several mitochondrial pathologies and during ageing.

## Results

### mtDNA alterations in skeletal muscle do not induce mitophagy

Expression of K320E in skeletal muscles drives differential accumulation of mitochondrial damage depending on fiber-type[23]. In extraocular muscles, mtDNA alterations preferentially accumulate in fast-twitch fibers in contrast to slow-oxidative fibers, where mitochondrial function is fundamental for muscle performance. We hypothesized that these differences reflected different mitochondrial quality control systems surveillant mitochondrial integrity in different muscle fiber types. To study the nature of these mechanisms, we first analyzed fast-twitch M. tibialis anterior (TA) and slow-oxidative M. soleus (SOL), muscles both rich in fibers with high mitochondrial content in mice, but with preferentially glycolytic (TA) vs. oxidative metabolism (SOL), respectively. As shown before, K320E mutant mice carry a wide variety of reorganized mtDNA molecules[27], causing an inefficient PCR amplification reaction and leading to a smear of products, however there were no changes in total mtDNA copy number (Fig. 1a, b). TA from aged WT mice also showed many mtDNA alterations, while only few were found in SOL. By conventional PCR, we analyzed the presence of common mtDNA deletions in aged mice[23] and selected a deletion covering about 4000 b.p (*Mus musculus* mtDNA-Δ[983-4977]), which was present in both mutant muscles (Supplementary Fig. 1a, ca. 500 bp product). Considering that mtDNA copy number is 20% higher in SOL than in TA (Supplementary Fig. 1b), we performed qPCR quantification using the D-Loop region as a reference and found that indeed, this deletion was on average 20 times more abundant in TA compared to SOL (Fig. 1c).

Steady-state protein levels of common mitochondrial autophagy markers showed that in SOL of K320E[SkM] mice, the general adaptors p62 and LC3-II were significantly decreased while their levels were similar in TA (Fig. 1d, e). In contrast, the specific mitophagy adaptor Optineurin (OPT) and the mitochondrial marker TOM20 were similar in all samples. In situ immunofluorescence of LC3 and p62 confirmed reduced number of puncta per fiber only in SOL (Fig. 1f, g; Supplementary Fig. 1c–e). Since such a reduction could mirror both an increase or a decrease of autophagic flux, we used chloroquine to block lysosomal acidification, and analyzed LC3 protein level, a method to monitor autophagy flux[28]. As expected, after autophagy block, we observed an accumulation of the autophagosome isoform LC3-II but only in control mice. However, in mutant mice, the cytosolic form LC3-I was depleted, with only subtle changes in LC3-II. LC3-II/LC3-I ratio, indicative of autophagosome turnover, revealed an increased flux only in SOL for K320E mice (Fig. 1h, i), however, following a shift

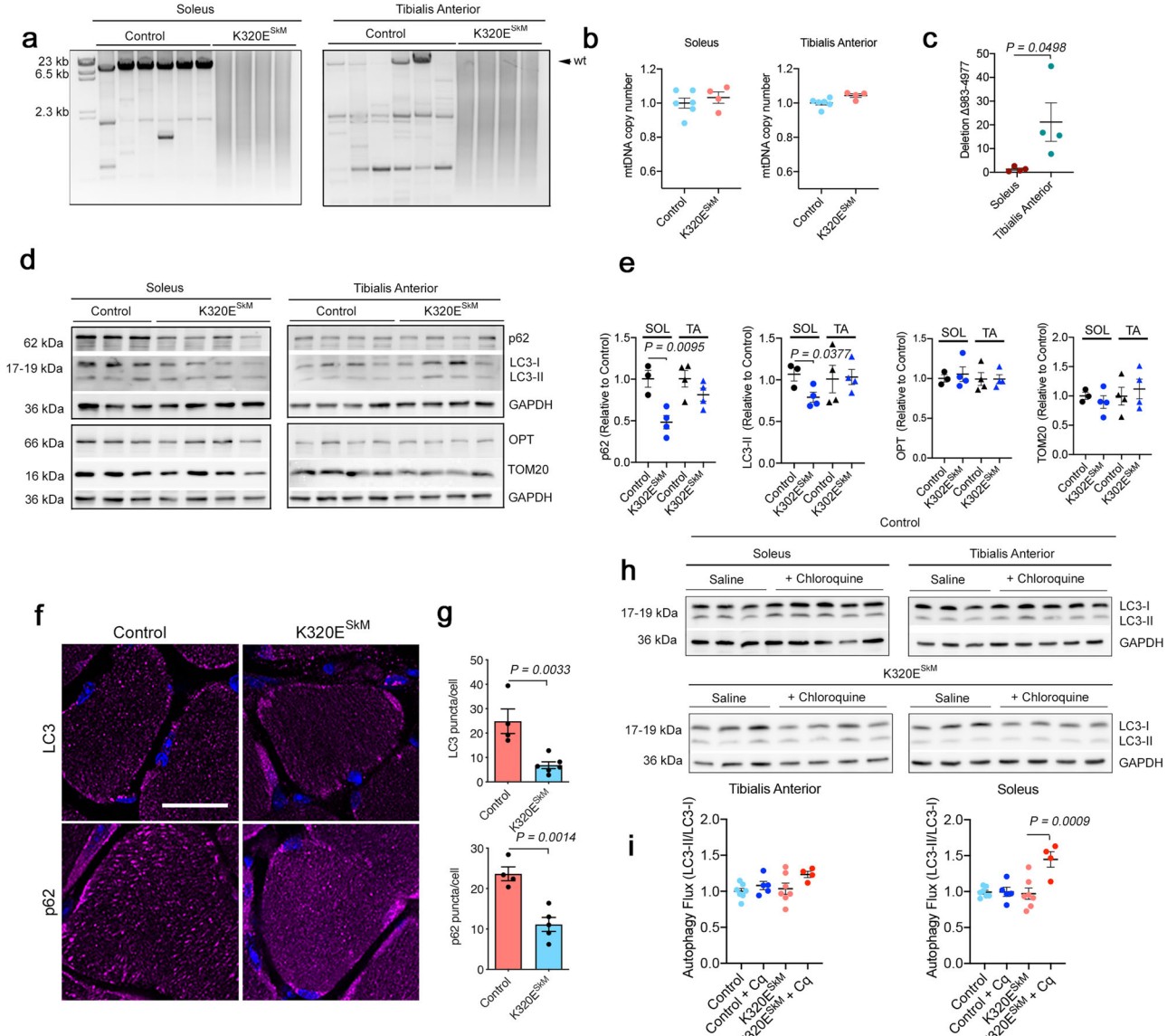

**Fig. 1 | In vivo expression of Twinkle-K320E induces differential accumulation of mtDNA alterations. a** Long range PCR analysis and **b** quantitation of mtDNA copy number in M. soleus and M. Tibialis anterior from 24 months old control and Twinkle-K320E$^{SkM}$ mice (control: $n = 6$; K320E: $n = 4$). **c** qPCR quantitation of deletion mtDNA-$\Delta$983-4977 in M. soleus and M. Tibialis anterior from 2 years old mice ($n = 4$). **d, e** Western blot analysis and quantification of the indicated proteins in muscle extracts from these mice. (Soleus, control: $n = 3$; Twinkle-K320E: $n = 4$. M. TA, control: $n = 4$; Twinkle-K320E: $n = 4$). **f, g** In situ immunofluorescence and image quantification showing autophagic markers LC3 and p62 in cryosections of M. soleus. 5 random pictures with 4 fibers per plane were analyzed per animal to obtain averaged values (control: $n = 4$; K320E: $n = 5$). Scale bar, 20 $\mu$m. **h, i** Autophagic flux analysis (LC3-II/LC3-I ratio) in muscle extracts from mice treated with saline or 50 mg/kg chloroquine (Cq) 4 hours before euthanisation. (control, $n = 7$; control +Cq, $n = 5$; K320E, $n = 7$; K320E + Cq, $n = 4$). $P$ values calculated using unpaired two-tailed Student's $t$ test (**c, e**), or One-way ANOVA with Tukey correction for multiple comparison (**h**). Data are presented as Mean $\pm$ SEM.

not supported by canonical forms of autophagy, where LC3-II accumulates after lipidation and conversion from LC3-I.

We hypothesized that oxidative fibers, which are most dependent on intact mitochondria, possess a faster mitochondrial turnover, thus maintaining mutated mtDNA molecules more efficiently below a pathogenic threshold. As mitochondrial turnover is a combination of mitochondrial biogenesis and mitophagy, we set up two different approaches to differentiate both pathways. In the first approach, we analyzed protein levels for different mitochondrial proteins in the steady state and after blocking autophagic flux in order to determine mitochondrial removal rate (Supplementary Fig. 2a–c). Unexpectedly, mitochondrial inner membrane proteins MIC19, ATP5A, UQCRC2, MTCO1, SDHB, NDUF8B, or matrix LRPPRC showed non-significant accumulation upon the degradation block. However, we observed a

significant enrichment of the mitochondrial outer membrane proteins TOM20 and TOM40 in SOL for K320E mice.

On the other hand, to visualize protein synthesis in vivo, we fed our mice with $^{13}C_6$-lysine for two weeks and analyzed heavy lysine incorporation, to get an estimate for mitochondrial biogenesis rates[29]. Here, we selected SOL and M. extensor digitorum longus (EDL) for analysis, since the latter contains almost exclusively fast glycolytic fibers. We confirmed that heavy lysine incorporation was equal in all animals (Supplementary Fig. 2d, e). We analyzed H/L (heavy/light) ratios of detected proteins (1286 and 1224 for SOL, 1042 and 1018 for EDL in wild type and K320E$^{SkM}$ mice, respectively) and filtered the mitochondrial proteins using MitoCarta 3.0 (Supplementary Data 1; Supplementary Fig. 2f). We observed no significant incorporation of heavy lysine in any mitochondrial protein in K320E animals, neither in

EDL nor in SOL, indicating the same rate of mitochondrial biogenesis (Supplementary Fig. 2g). All these data together suggest that, in muscles bearing mtDNA alterations, mitochondrial turnover is not enhanced, as we could observe neither an accumulation of mitochondrial matrix or inner membrane proteins, nor an increase in mitochondrial biogenesis. Mitochondrial outer membrane proteins however might be subjected to a faster turnover.

## Lysosomal function is required for mtDNA depletion following damage

In order to further examine molecular pathways activated upon mtDNA instability, we generated stable C2C12 myoblast cell lines expressing tagged versions of Twinkle and K320E. Colocalization of these variants with mitochondria (outer membrane marker TOM20, Supplementary Fig. 3a, b) as well as with mtDNA and the mtDNA binding protein TFAM was verified (Supplementary Fig. 3c, d), confirming that transgenic Twinkle is enriched in mitochondrial nucleoids.

To investigate if K320E expression leads to activation of autophagy or mitophagy, C2C12 myoblasts were additionally transfected with tandem plasmids encoding LC3-GFP-mCherry or Fis1p-GFP-mCherry. In both cases, the acidic pH of phagosomes, after fusion with lysosomes, quenches GFP fluorescence, facilitating visualization and quantification of autophagy or mitophagy flux[28]. Thus, expression of K320E induced the accumulation of autolysosomes marked by LC3-GFP-mCherry (Fig. 2a, c; magenta signal) but, in agreement with our in vivo results, we could not observe activation of bulk mitophagy (Fig. 2b, d). Previous studies have demonstrated that in vitro expression of several Twinkle missense mutations interferes with mtDNA replication inducing mtDNA depletion[30]. Consistently, expression of K320E in our cells led to mtDNA depletion (Fig. 2e), although this was not related to a decrease in mtDNA replication rate, as observed by analysis of mtDNA replication foci in BrdU labeled cells (Supplementary Fig. 4a, b). Hence, we hypothesized that the accumulation of lysosomes and mtDNA depletion after expressing K320E reflects a selective mtDNA degradation process. Analysis of the spatial relationship between Twinkle foci and autophagy structures, using LC3 as an autophagosome marker and LAMP1 as a late endosomal marker, showed K320E foci colocalizing with LC3 and LAMP1 (Fig. 2f, g), albeit with most of the Twinkle foci distributed in a pattern decorating the mitochondrial network. In addition, while chloroquine treatment was sufficient to recover mtDNA copy number to control levels, blocking of autophagy initiation using 3-MA or SBI0206965 was ineffective (Fig. 2h), suggesting that lysosomes, but not canonical autophagy, is related to mtDNA removal upon mtDNA instability.

Our data show that K320E expression increases mtDNA degradation and therefore, that K320E affects mtDNA, however by unknown means. To visualize in situ mtDNA damage, we examined the presence of 8-hydroxy-2'deoxyguanosine (8-OHdG), a specific base modification induced by reactive oxygen species[31]. In the steady state, we could not observe any specific staining with α−8-OHdG (Supplementary Fig. 4c, d), however, when cells were incubated in the presence of chloroquine, we detected a marked accumulation of 8-OHdG decorating the mitochondrial network in K320E cells (Fig. 2i, j).

To further analyze the molecular players for specific mtDNA turnover, we turned to *Atg5* knockout MEFs and expressed Twinkle variants (Fig. 3a). Atg5 has been shown to be important in quality control after mitochondrial damage and has been involved in both canonical and non-canonical autophagy[32,33]. Analysis of mitochondrial nucleoids showed that K320E expression reduces nucleoid foci number only in *Atg5* WT cells (Fig. 3b) but, paradoxically, mtDNA copy number was reduced in both *Atg5* WT and *Atg5* KO cells (Fig. 3c). Analysis of mtDNA replication by BrdU labeling showed that, in contrast to WT cells, expression of K320E in ATG5 deficient cells led to a reduced number of replicating mtDNA foci within the mitochondrial network (Fig. 3d, e). Consistent with our previous observation,

chloroquine treatment, but not autophagy inhibitors, restored mtDNA copy number in K320E-*Atg5* WT cells but not in K320E-*Atg5* KO cells (Fig. 3f, g). Our data suggest that in ATG5 deficient cells, mtDNA depletion is caused by reduced mtDNA replication, presumably to avoid accumulation of excessive mtDNA damage. In contrast, in WT cells, K320E triggers mtDNA depletion linked to an increased turnover rate.

To evaluate the role of ATG5 in mtDNA depletion, we looked for specific means to damage the mtDNA in a manner similar to the damaged caused by the expression of K320E. We selected CCCP as a general mitochondrial uncoupler, and Ethidium bromide (EtBr) and $H_2O_2$ as specific mtDNA stressors and analyzed their effect on several mitochondrial parameters. First, we quantified changes in mitochondrial morphology (Supplementary Fig. 5a, b) and observed that, while CCCP and $H_2O_2$ induced fragmentation of the mitochondria, neither EtBr nor K320E affected the mitochondrial network. mtDNA damage quantified by Long Run PCR amplification[34] confirmed that the relative number of lesions in the mtDNA were increased in cells treated with EtBr or $H_2O_2$, and in cells expressing K320E (Supplementary Fig. 5c). Finally, we checked whether any of these treatments affected the stoichiometry of OXPHOS subunits and observed that, while EtBr and K320E only affected the mtDNA encoded OXPHOS subunits (NDUF8B, complex I; UQCRC2, complex III; MTCO1, complex IV), CCCP or $H_2O_2$ did not alter their protein level (Supplementary Fig. 5d–i). Similar to K320E expression, EtBr also induces accumulation of 8-OHdG within the mitochondrial network (Supplementary Fig. 4e, f), without interfering with mtDNA replication, at least for the duration of our experiment (Supplementary Fig. 4g, h). All these data together suggest that short treatment with EtBr damages mtDNA, and can be used as a chemical means to interfere with mtDNA structure in a similar fashion to K320E expression.

Consequently, we treated ATG5 deficient cells with EtBr and analyzed the effect on mtDNA and mitochondrial nucleoids. EtBr decreased mtDNA foci number in *Atg5* WT but not in *Atg5* KO cells (Fig. 3h, i). In line with this, *Atg5* control cells showed consistent mtDNA depletion, whereas in *Atg5* KO cells mtDNA copy number remained unchanged (Fig. 3j). EtBr-induced depletion could not be recovered by overexpression of WT Twinkle, indicating a prominent role of ATG5 in mtDNA-induced depletion. Nevertheless, we wondered if mtDNA depletion induced by EtBr may alternatively reflect a reduced mitochondrial biogenesis rate. Hence, we checked expression of *Pgc1α* mRNA, a well-known master regulator for mitochondrial biogenesis, and found that, instead, EtBr upregulated *Pgc1α* mRNA levels (Supplementary Fig. 4i), validating that the effect of EtBr, in our experimental approach, was related to an increased mtDNA degradation. With these data, we confirm that mtDNA damage induces activation of an mtDNA turnover mechanism, which relies on lysosomal activity and ATG5, and specifically degrades mtDNA independent of canonical autophagy or mitophagy.

## MtDNA damage induces local reorganization of mitochondrial membranes

Recent research has demonstrated that elimination of mutated mtDNA in yeast depends on the sphere of influence of particular nucleoids containing mutated mtDNA[35]. Expression of K320E induces accumulation of mtDNA damage, either by generating mtDNA alterations[21] (Fig. 1a), as well as specific oxidative damage (Fig. 2i, j). Hence, cells expressing this variant of the mitochondrial helicase might carry an abundant number of nucleoids bearing mtDNA alterations. To detect proteins involved in the enhanced mtDNA turnover of K320E cells, we generated cells stably expressing APEX2 fused with Twinkle and K320E. In a short $H_2O_2$ pulse, APEX2 catalyzes the biotinylation of proteins in its proximity, which can be either detected by immunofluorescence or further purified and analyzed by mass spectrometry, facilitating the identification of neighboring proteins[36].

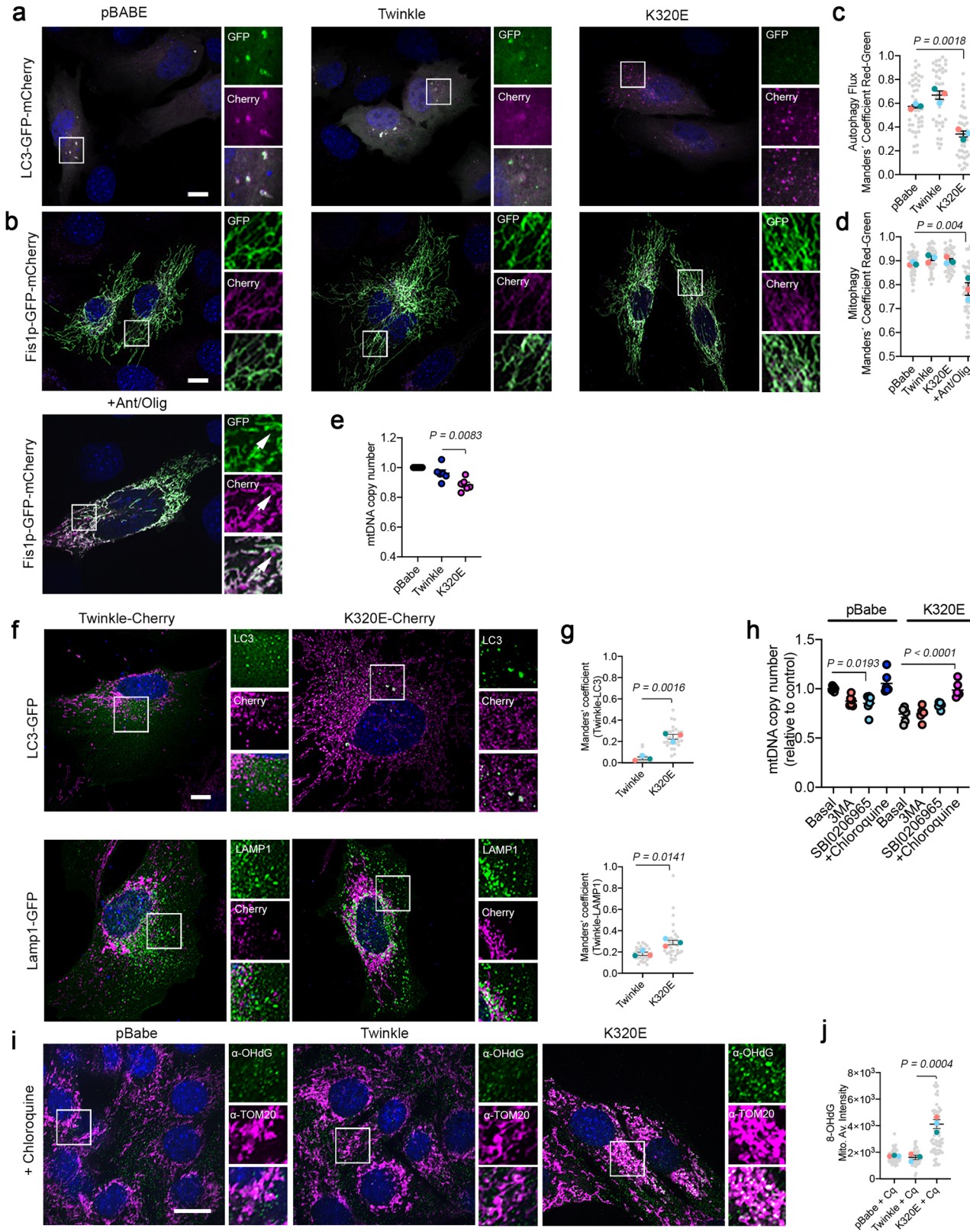

Thus, we performed Twinkle-APEX2 and K320E-APEX2 proximity labeling to detect specifically nucleoid neighboring proteins. Immunofluorescence and colocalization of biotinylated proteins with TOM20 confirmed that the reaction was specific for the mitochondrial compartment, as observed for matrix targeted APEX2 (mitoAPEX2) control cells and Twinkle-APEX2 cells (Fig. 4a; Supplementary Fig. 6a, b). Next, we analyzed enriched proteins following APEX2 biotinylation

by mass spectrometry. Analysis of the proximity proteomes revealed that, as expected, the majority of the enriched proteins were annotated in MitoCarta 3.0, with greater presence of proteins related to vesicle trafficking and endosomes in K320E-APEX2 (Fig. 4b; Supplementary Data 2). To detect specific proteins in the neighboring area of the nucleoids, we filtered for mitochondrial targets significantly enriched compared to the mitoAPEX2 control, and significantly regulated

**Fig. 2 | Twinkle-K320E triggers mtDNA damage and induces autolysosome accumulation independent of canonical mitophagy. a** C2C12 expressing untagged Twinkle constructs transiently expressing the autophagy reporter LC3-GFP-mCherry and **b** the mitophagy reporter Fis1p-GFP-mCherry. Red signal shows lysosomal localization. Arrows indicate mito-lysosomes. **c, d** Manders' colocalization coefficient Red/Green quantification of transfected cells. A decrease in Manders' coefficient indicates autolysosome or mito-lysosome accumulation. Cells treated overnight with 10 μM Antimycin/Oligomycin were used to induce canonical mitophagy (n = 3, 10–15 transfected cells per replicate). **e** mtDNA copy number in C2C12 cells stably expressing Twinkle and Twinkle-K320E (K320E), respectively, vs. empty vector (pBABE). (n = 6 independent cultures). **f** Confocal images and quantification of C2C12 expressing mCherry tagged Twinkle transfected with plasmids encoding the autophagosome marker LC3-GFP and lysosome marker Lamp1-GFP. **g** Manders' colocalization coefficient was used to confirm Twinkle colocalization with autophagic organelles (n = 3, 10–15 transfected cells per replicate). **h** mtDNA copy number in cells treated with the autophagy inhibitor 3MA, SBI0206965, or the lysosomal inhibitor chloroquine for 24 h (n = 6 independent cultures). **i** mtDNA oxidative damage detected by immunofluorescence with α-OHdG and α-TOM20 antibodies in cells expressing Twinkle variants and treated with chloroquine for 24 h. **j** Relative intensity quantification of the α-OHdG signal inside the mitochondrial network. (n = 3, >20 cells per replicate). Scale bar, 10 μm. P values were calculated using unpaired two-tailed Student's t test (**g**), or one-way ANOVA with Tukey correction for multiple comparison (**c, d, e, h,** and **j**). Data are presented as Mean ± SEM.

between Twinkle-WT and Twinkle-K320E. We then Z-score-normalized log2 LFQ intensities and performed Euclidean hierarchical clustering. Using this approach, we observed the presence of specific mitochondrial inner and outer membrane proteins for K320E which were not detected in the neighboring proteome of WT Twinkle (Fig. 4c–e; Supplementary Fig. 6c). The enrichment of mitochondrial inner and outer membrane proteins in K320E cells was further proven by gene ontology analysis (qualitative: Fisher exact test; quantitative: 1D annotation enrichment, for details see Supplementary Data 2). Among them, proteins such Prohibitin2, SAMM50, VDAC1, VDAC2 and VDAC3 were enriched, targets related to membrane architecture, protein transport and pore formation (Fig. 4e). To detect further mitochondrial targets hidden by comparisons with mitoAPEX2, we performed a second analysis but using the empty vector pBabe for statistics (Fig. 4f; Supplementary Data 2). Again, we confirmed SAMM50 enriched in K320E samples and detected new mitochondrial inner membrane proteins like ATAD3. Interestingly, we found ATAD3 also to be an interactor of Twinkle by direct IP followed by MS analysis (Supplementary Fig. 6d; Supplementary Data 3).

The differential proteome detected for Twinkle and K320E suggest that, as observed in yeast[35], the sphere of influence of the mitochondrial nucleoid changes depending on the status of the mtDNA, probably modifying the local membrane architecture. Indeed, ATAD3 enrichment after Twinkle pull-down was modified by the presence of K320E, and SAMM50 was strongly represented when the pull-down proteome of Twinkle and K320E was compared (Supplementary Fig. 6d; Supplementary Data 3). In agreement with the proteomics data, Twinkle pull-down experiments showed that the association of Twinkle with SAMM50 and ATAD3 depends on the status of the mtDNA. While ATAD3 was found to immunoprecipitate with Twinkle in steady state, the interaction was lost upon mtDNA damage with EtBr (Fig. 4g). On the contrary, SAMM50 was found to precipitate with WT Twinkle only upon EtBr mtDNA damage (Fig. 4h) and consistently, also with K320E in steady state (Supplementary Fig. 6e).

ATAD3 and SAMM50 have been found to be part of a big mitochondrial protein complex containing several inner and outer membrane proteins and regulating mitochondrial inner and outer contact sites[37] (Supplementary Fig. 6f). ATAD3 is a transmembrane protein which crosses the mitochondrial inner membrane but also contacts the outer membrane. The Nt region of ATAD3 is required for the interaction with the inner surface of the mitochondrial outer membrane (aa 2–49), the intermediate part crosses the MIM and, the Ct region faces the mitochondrial matrix where it binds to mitochondrial nucleoids[38]. To determine if ATAD3 was binding directly to SAMM50 through the intermembrane space domain (IMS), we generated cells expressing HA-tagged constructs of ATAD3 (ATAD3-HA) and deletion of IMS domain (ΔIMS-ATAD3-HA) (Supplementary Fig. 6g). Co-immunoprecipitation experiments confirmed that ATAD3-SAMM50 interaction sustains even after removing the IMS domain (Fig. 4i), suggesting that their association is through the membrane domains. In addition, we found that EtBr disrupts ATAD3-SAMM50 interaction but not ATAD3 binding to the inner mitochondrial membrane protein IMMT/MIC60 (Fig. 4j), suggesting that mitochondrial inner and outer membrane contact sites are disrupted.

To visualize mitochondrial membrane changes incited by specific mtDNA damage, we analyzed mitochondrial cristae morphology using STED microscopy and PK Mito Orange staining. Super resolution images revealed that mitochondria cristae structure was dramatically altered upon EtBr treatment (Supplementary Fig. 7a, b). Similar changes were observed in cells expressing K320E (Supplementary Fig. 7a, b). Further, TMRE staining and super resolution microscopy by Airy Scan showed that EtBr treatment induces a more extensive heterogeneity of the local mitochondrial membrane potential, depicting mitochondrial network parts, with a shift to very low (blue) and low potential (green) (Supplementary Fig. 7c). However, bulk mitochondrial membrane potential quantified by Flow cytometry revealed only subtle changes at a cellular level (Supplementary Fig. 7e). K320E expressing cells showed similar results but with a significant reduction of the bulk membrane potential (Supplementary Fig. 7d, f).

**The ATAD3-SAMM50 axis is necessary for mtDNA extraction**

Our data suggests that mitochondrial inner and outer membrane contact sites are disrupted upon mtDNA damage, inducing local changes in membrane potential and cristae morphology, which could initiate nucleoid extraction and elimination. To ascertain the specific role of SAMM50 and ATAD3 in mtDNA turnover, we generated constitutive Atad3 and Samm50 mRNA knock-down (KD) clones (Supplementary Fig. 8a, b) and studied mtDNA dynamics upon mtDNA damage. Interestingly, KD of one protein interfered with steady state levels of the other partner (Supplementary Fig. 8c). Nevertheless, neither ATAD3 nor SAMM50 depletion affected severely mitochondrial morphology, membrane potential or cristae morphology in steady state (Supplementary Fig. 8d–f). We then treated the cells with EtBr to induce mtDNA depletion following damage and quantified mtDNA foci. For Atad3 KD cells, we observed that upon EtBr treatment, mtDNA foci were preserved inside mitochondria (Fig. 5a, b) but were shifted to a bigger size (Supplementary Fig. 8g, h). mtDNA copy number analysis confirmed these results (Fig. 5c). Nucleoid aggregation and mtDNA copy number were recovered upon re-expression of ATAD3-HA and ΔIMS-ATAD3 (Fig. 5d; Supplementary Fig. 8i, j).

On the other hand, in Samm50 KD cells, EtBr treatment reduces mtDNA foci number inside the mitochondrial compartment (Fig. 5a, e). Intriguingly, we detected an increased number of mtDNA foci remaining outside the mitochondrial network which was incremented upon EtBr treatment (Fig. 5f). mtDNA copy number was also reduced by EtBr treatment (Fig. 5g). mtDNA release upon mitochondrial damage has been described to occur through BAK/BAX pore oligomerization[39]. This process is linked to the permeabilization of the mitochondrial membranes allowing release of mitochondrial components[40]. Interestingly, SAMM50 has been found to interact with BAK[37], a feature which we could also replicate (Supplementary Fig. 8k). Different protein domains included in SAMM50 are responsible for its distinctive functions[41]. Thus, the Nt cytosolic domain (aa 1–40) is required for basal mitophagy, the IMS POTRA domain (aa 45–125) is necessary for protein interaction with

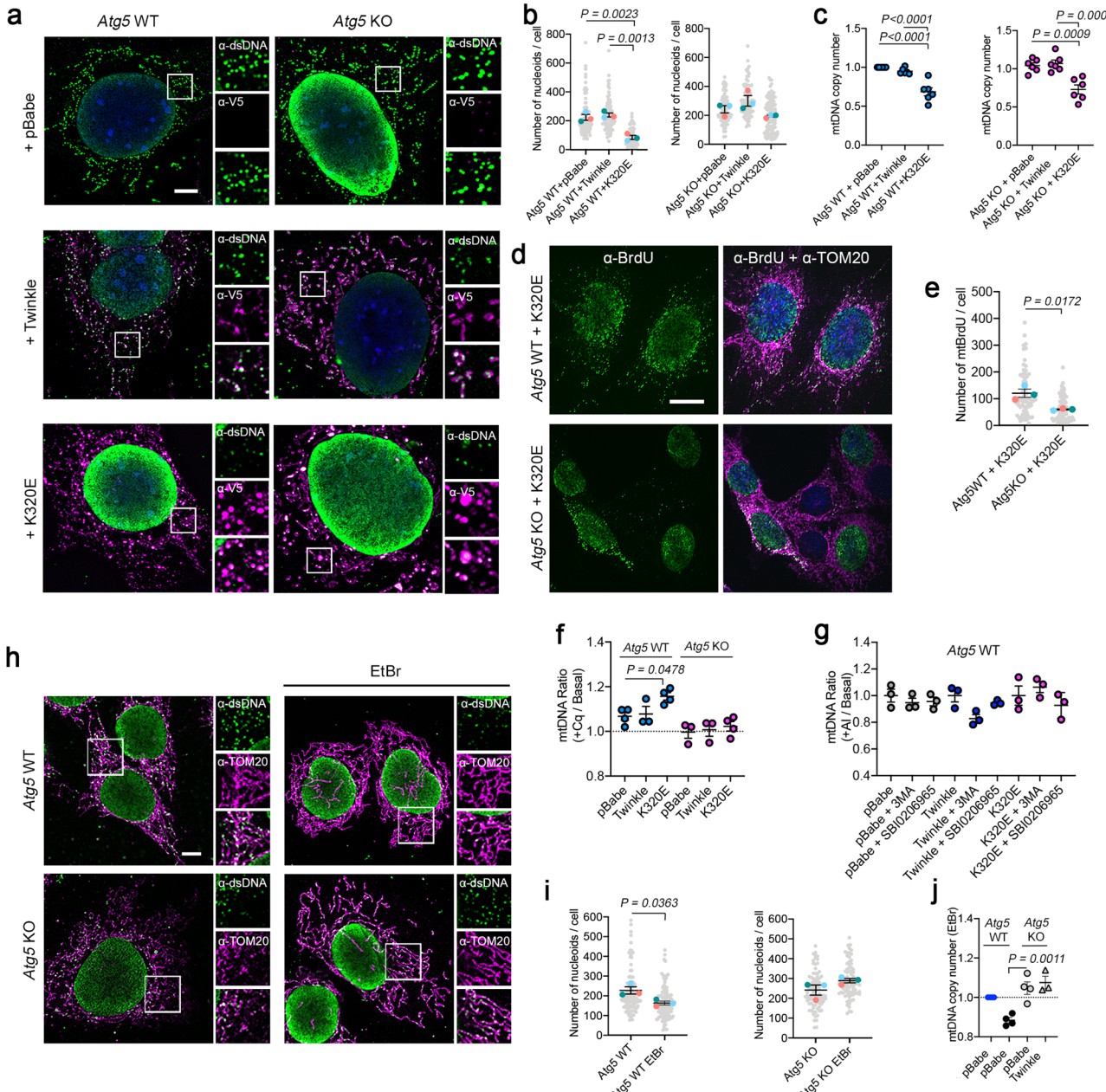

**Fig. 3 | ATG5 is required for mtDNA clearance after mtDNA damage.** *Atg5* WT and KO cells were transduced with Twinkle-APEX2-V5 plasmids. **a** α-DNA and α-V5-tag immunofluorescence confirming localization of Twinkle in mtDNA nucleoids. **b** Quantification of mtDNA foci number in *Atg5* WT and *Atg5* KO cells (*n* = 3, >25 cells per experiment). (**c**) Steady state mtDNA copy number in *Atg5* cells (*n* = 6 independent cultures). **d, e** α-BrdU and α-TOM20 immunofluorescence and quantification of mtDNA replication foci detected by treating the cells for 6 h with 20 μM BrdU. (*n* = 3, >20 cells per replicate). **f** Quantification of mtDNA copy number in *Atg5* cells treated with Chloroquine or **g** the autophagy inhibitors 3MA and

SBI0206965 for 24 h (*n* = 3–4 independent cultures). Ratio between the treated and basal value was performed after normalization with internal controls. **h, i** α-DNA and α-TOM20 immunofluorescence and mtDNA foci quantification in steady state and in cells treated with 50 ng/ul EtBr for 7 days. (*n* = 3, >20 cells per replicate). **j** mtDNA copy number analysis for steady state and cells treated for 5 days with EtBr (*n* = 3–4 independent cultures). Scale bar 5 μm (**a**), or 10 μm (**d** and **h**). *P* values calculated using unpaired two-tailed Student's t test (**e** and **i**), or One-way ANOVA with Tukey correction for multiple comparison (**b, c, f, g,** and **j**). Data are presented as Mean ± SEM.

mitochondrial inner membrane proteins such as MIC19, while the β-barrel domain (aa 151–469) is linked to protein insertion and translocation into the mitochondrial membrane[41] (Supplementary Fig. 6f). To shed light on the involvement of SAMM50 in mtDNA release through the BAX/BAK pore, we generated cells expressing HA-tag versions of the wt protein and a deletion construct lacking the Nt domains (ΔPOTRA-SAMM50) (Supplementary Fig. 8l). First, we checked for mitochondrial distribution of BAK and found that in steady state BAK had a diffuse pattern decorating the mitochondrial network and colocalizing with SAMM50 (Fig. 5h). Interestingly, EtBr treatment induced BAK clustering

and disrupted SAMM50-BAK overlapping. No colocalization was observed for ΔPOTRA-SAMM50 neither in steady state or upon EtBr (Fig. 5i). In agreement with an inhibitory role of SAMM50 towards BAK[37], we found elevated levels of transcription factors related to the innate immune response in *Samm50* KD cells, which are activated upon increased levels of mtDNA in the cytoplasm[42] (Fig. 5j). These data suggest that ATAD3 and SAMM50 have complementary roles in mtDNA turnover. mtDNA damage alters the distribution of mitochondrial membranes, affecting the ATAD3-SAMM50 axis, and activates a signaling cascade involved in mtDNA release.

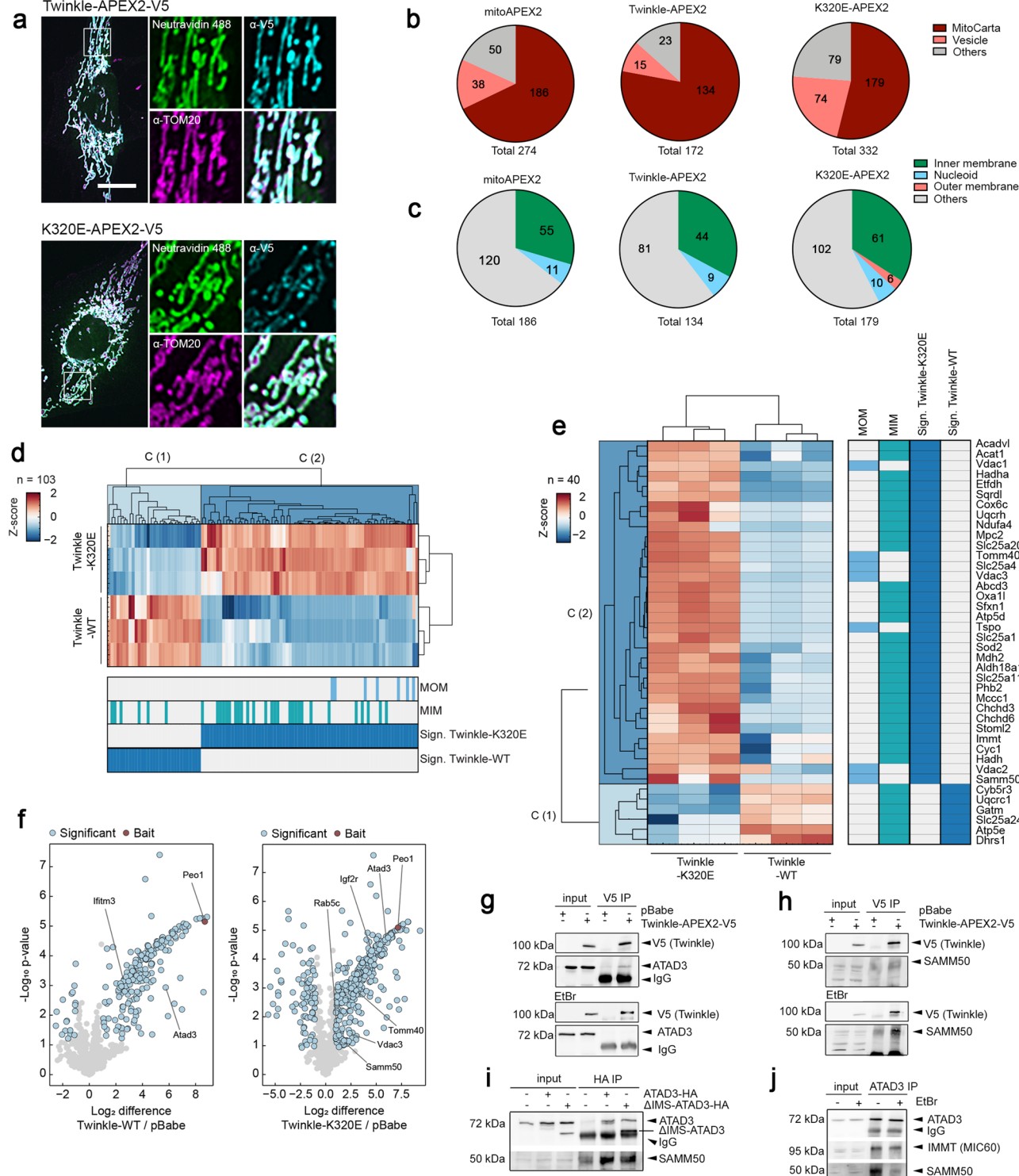

**Fig. 4 | Mitochondrial membranes reorganize upon mtDNA damage.**
**a** Immunofluorescence of C2C12 cells expressing Twinkle-APEX2-V5 variants exposed to biotin-phenol crosslinking. Biotinylated proteins were detected with α-Neutravidin, mitochondria with α-TOM20 and α-V5 for Twinkle. Scale bar 10 µm. **b** Pie charts of significant hits detected by MS after biotin-phenol crosslink (significance = q-value <0.05 and absolute log2 difference >1). Groups were created according to MitoCarta and GOCC annotation. **c** Pie charts showing all significantly enriched MitoCarta annotated proteins and their mitochondrial sub-compartment. **d** Hierarchical clustering of Z-score-normalized protein targets. Enrichment of proteins annotated in MitoCarta and, **e** with proteins previously described to be located in the mitochondria inner (MIM) or outer membrane (MOM) for Twinkle

and K320E. Targets were selected based on significant enrichments compared to mitoAPEX2 in the matrix (significance: q-value <0.05 and absolute log2 difference >1). **f** Volcano plots showing proteins enriched after crosslinking and purification of Twinkle and K320E-APEX2-V5. Differentially enriched proteins compared with cells transfected with empty vector pBabe (significant: q value < −0.05 and absolute log2 fold change >1) are highlighted in blue. Twinkle (*Peo1*) is highlighted in red. **g** Co-immunoprecipitation of V5-tagged Twinkle and ATAD3 or **h** SAMM50, in steady state or after EtBr treatment. **i** Co-immunoprecipitation of HA-tagged ATAD3 and ΔIMS-ATAD3 with SAMM50. **j** Co-immunoprecipitation of ATAD3, SAMM50 and IMMT/MIC60 in steady state or after EtBr treatment.

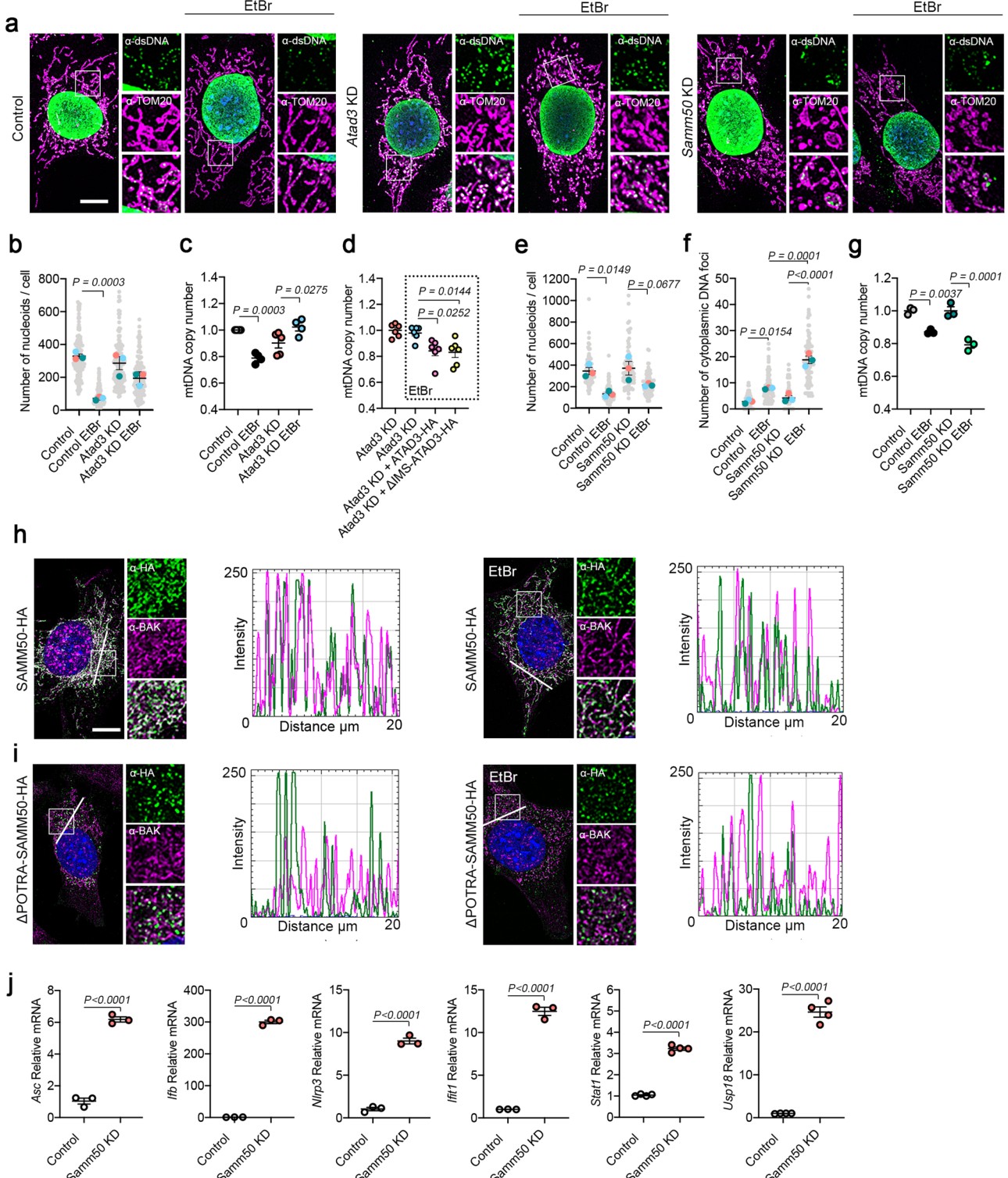

**Fig. 5 | ATAD3 and SAMM50 coordinate for mitochondrial nucleoid extraction.**
**a** α-TOM20 and α-dsDNA immunofluorescence in *Atad3* and *Samm50* KD MEFs in steady state and after EtBr treatment. **b** mtDNA foci quantification in *Atad3* KD cells. Only the dsDNA signal in contact with the mitochondrial marker TOM20 was considered (*n* = 3, >40 cells per replicate). **c** mtDNA copy number quantification after *Atad3* KD and **d** after transduction of ATAD3-HA or ΔIMS-ATAD3-HA and EtBr treatment (*n* = 6 independent cultures). **e** mtDNA foci quantification after *Samm50 KD* in steady state and after EtBr treatment inside the mitochondrial matrix and **f** located in the cytosol. (*n* = 3, >20 cells per replicate). **g** mtDNA copy number

analysis for *Samm50 KD* cells (n = 3 independent cultures). **h** Representative images of α-BAK and α-HA immunofluorescence in cells transduced with SAMM50-HA and (**i**) ΔPOTRA-SAMM50-HA. Fluorescence intensity profiles were obtained for a 20 μm straight line crossing the cell body. **j** mRNA quantification of pro-inflammatory genes in *Samm50 KD* cells activated in response to cytosolic mtDNA. (*n* = 3–4 independent cultures). Scale bar, 10 μm. *P* values calculated using unpaired two-tailed Student's t-test (**j**), or one-way ANOVA with Tukey correction for multiple comparison (**b**, **c**, **d**, **e**, **f**, and **g**). Data is presented as mean ± SEM.

## mtDNA removal requires coordination of endosomes and the retromer

To visualize nucleoids containing mtDNA mutations in their cellular context, we took advantage of APEX2 ability to induce 3,3′-Diaminobenzidine (DAB) deposition and provide contrast for electron microscopy[43]. Using this approach, we observed that K320E-APEX2 accumulated in poles of the mitochondrial network, sometimes close to abnormal membrane structures (Fig. 6a, arrows). To identify the nature of these membranes, we performed electron tomography and 3D reconstruction of mitochondria presenting these abnormal structures (Fig. 6b). Following mitochondrial membranes through the different planes of the stack, we identified MIM (green) and MOM (magenta). We also identified other membranes that could not be attributed to the mitochondria and originated from nearby organelles (blue).

The abnormal mitochondrial structures in cells carrying mtDNA alterations and the presence of a high number of vesicle related proteins in the proximity proteome for K320E cells (Fig. 4b), prompted us to investigate it in greater detail. Following the same approach as used for mitochondrial membrane proteins, we performed a hierarchical clustering for vesicle related and endo/lysosomal proteins (Fig. 6c). When we compared enrichments only for endo/lysosomal proteins, we detected up to 15 targets significantly enriched in cells expressing K320E-APEX2 compared to mitoAPEX2, including the early endosome marker RAB5C, the vesicle trafficking protein FYCO1, and the lysosomal enzymes LGMN, CTSB, HEXA and GUSB, while only 3 of these proteins were enriched in WT expressing cells (Fig. 6d; Supplementary Fig. 9a).

Our data evidence the role of the endo-lysosomal pathway in mtDNA removal upon mtDNA damage. Some endosomal populations containing RAB5 (early endosomes) and RAB7 (late endosomes) are linked to non-canonical forms of mitophagy[10,44]. In addition, VPS35, the main component of the endosomal retromer complex, is involved in maturation of endosomes[45] and has been linked to mitophagy through MDVs[5], where it initiates the force to generate a vesicle[7]. Therefore, we checked if mitochondria-RAB5 contact sites were affected in presence of mtDNA alterations, however we could not identify any change in K320E cells (Supplementary Fig. 9b, c). Nonetheless, in cells bearing mtDNA alterations, we noticed an increased proportion of RAB5 early endosomes in contact with the retromer component VPS35 (Fig. 6e, f). Then, we checked for VPS35 colocalization with mitochondrial nucleoids and mitochondria, and observed that the percentage of VPS35 foci containing Twinkle but excluding TOM20 was increased in K320E cells and strongly upregulated, also in wt-Twinkle expressing cells, upon EtBr treatment (Fig. 6g, h). The overall steady state levels of VPS35 structures did not change (Fig. 6i). Importantly, the number of VPS35 foci containing dsDNA was increased as well in K320E expressing cells (Supplementary Fig. 9d, e). Moreover, in K320E cells, VPS35 accumulates after lysosomal degradation block with chloroquine without interfering with late endosomes or mitochondrial content (Supplementary Fig. 9f, g).

Despite being an endosomal related protein, lack-of-function mutation of VPS35 has a direct effect on mitochondrial homeostasis[46]. Thus, we generated Vps35 KO clones (Vps35_KO: Exon 4 and Vps35_KO: Exon 5; Supplementary Fig. 9h, i) and analyzed the effect on mitochondria and mtDNA. Mitochondrial morphology analysis revealed that Vps35 KO cells showed mitochondrial fragmentation already in steady state (Fig. 7a, b) and mtDNA depletion (Fig. 7c), suggesting an increased mitochondrial turnover. Thus, we expressed Fis1p-GFP-mCherry and found that, indeed, in Vps35 KO cells, bulk mitophagy was exacerbated (Fig. 7d, e).

To ascertain if VPS35 role in mtDNA turnover was linked to the MDV machinery we studied colocalization of Twinkle and VPS35 after expressing MAPL-GFP, knowing that MAPL has been shown to induce MDV formation[5]. Hence, consistent with previous studies, overexpression of MAPL induced mitochondrial fission and formation of MDVs excluding TOM20 but, importantly, these particles also excluded Twinkle (Supplementary Fig. 10a). Consistently with our previous results, the formation of Twinkle particles independent of TOM20 and MAPL was incremented upon chloroquine treatment (Supplementary Fig. 10b). Conversely, we found that VPS35 colocalized with MAPL but upon chloroquine treatment shifted to Twinkle particles mainly in K320E cells (Supplementary Fig. 10c−e). These results confirm that mtDNA turnover is independent of the MDV protein MAPL.

## SAMM50 is necessary to deliver nucleoids to VPS35 endosomes

VPS35 and SAMM50 have been shown to interact with the PINK1-Parkin pathway[8,47], and SAMM50 has been shown to be involved in mitochondrial quality control[41]. We hypothesize that SAMM50, a protein located in the mitochondrial outer membrane, could serve as a platform to recruit VPS35 facilitating the elimination of damaged mitochondrial nucleoids.

Thus, we used Samm50 KD clones, expressed Twinkle-mCherry, and treated the cells with EtBr to analyze VPS35-mitochondrial recruitment (Fig. 7f). As observed before, EtBr did not affect the overall population of VPS35 but enhanced close contacts to Twinkle (Fig. 7g, h). In Samm50 KD cells, however, the percentage of VPS35-Twinkle contact sites, was unchanged. We wondered if VPS35 recruitment showed Twinkle specificity, hence we examined VPS35 contact sites with LRPPRC, a mitochondrial matrix protein (Supplementary Fig. 11a, b). In this case, we observed that VPS35-LRPPRC contacts were not increased upon EtBr treatment and therefore, no effect was observed following SAMM50 downregulation. In addition, we were able to pull-down SAMM50 together with VPS35 (Supplementary Fig. 11c). We have shown that upon EtBr treatment, dsDNA foci could be detected outside the mitochondrial compartment and this was exacerbated upon Samm50 KD (Fig. 5f). Noteworthy, we observed that, upon EtBr, mtDNA indeed localized indeed with VPS35 foci in control cells, but not in Samm50 KD cells (Fig. 7i, j).

We wondered if mtDNA damage triggers the formation of vesicles through VPS35. Hence, we investigated by high-resolution microscopy VPS35 localization upon mtDNA damage and performed EM immunogold staining. We observed that VPS35, regardless of the treatment, was always detected in cytoplasmic foci, but not directly on the mitochondrial membrane (Fig. 7k). To determine the identity of the VPS35-Twinkle structures, we performed Correlative-Light-Electron-Microscopy (CLEM), by transfecting Twinkle-Cherry cells with VPS35-GFP (Fig. 7l, Supplementary Fig. 11d). Consistently, with our previous data, in the steady state, VPS35 and Twinkle were localized in the cytoplasm and in mitochondria respectively. However, upon mtDNA damage, we observed that VPS35-Twinkle colocalization foci increased, and remarkably, prominent structures containing both VPS35 and Twinkle were without any doubt late autophagy organelles (Fig. 7l). Super-resolution microscopy showed that upon EtBr, VPS35 could be found in the same cellular structure with BAK and SAMM50, and BAK and dsDNA (Fig. 7m; Supplementary Fig. 11e, f). In contrast, ΔPOTRA-SAMM50 failed to colocalize with BAK or VPS35 (Supplementary Fig. 11g). All these data confirm that the role of VPS35 in mtDNA clearance is independent of vesicle formation. VPS35 participates in mtDNA quality control by recruiting an endosomal population in the vicinity of BAK where mitochondrial nucleoids are engulfed in endosomes. VPS35 participates in the maturation of these endosomes and their fusion with lysosomes or late autophagy structures.

## Rapamycin specifically eliminates mtDNA alterations in vivo

Stimulation of mitochondrial turnover has been shown to be beneficial against mitochondrial diseases. Rapamycin, an activator of autophagy and lysosomal biogenesis through the specific inhibition of mTORC1, is able to revert mitochondrial dysfunction and ameliorates disease in

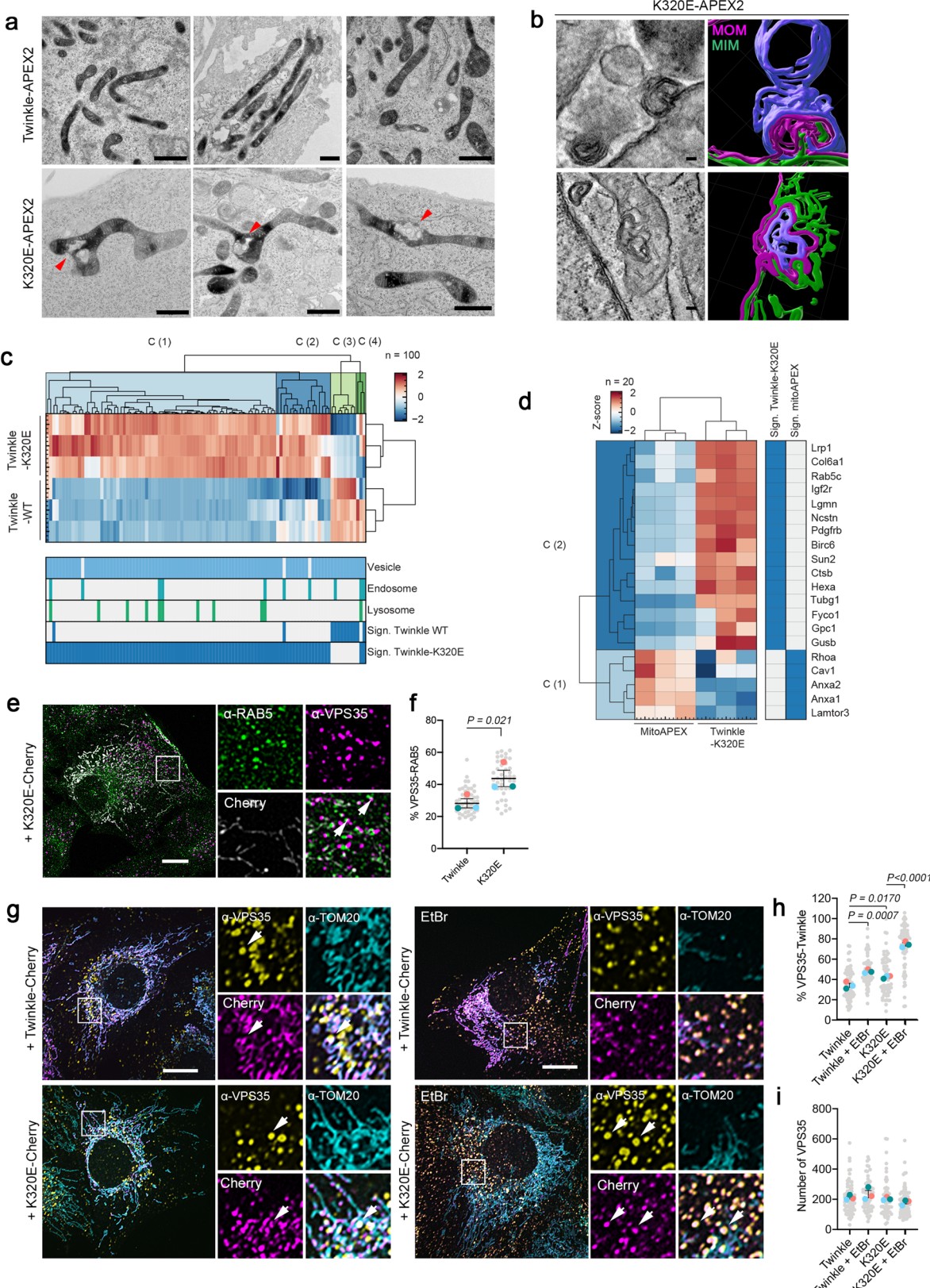

mice[48]. Modulation of autophagy has been reported to direct selection against mtDNA mutations in vitro and in a *Drosophila* model in vivo[49,50]. Since our data highlight a specific nucleoid extraction mechanism linked to the endo-lysosomal pathway playing an important role in maintaining mtDNA fitness in vitro, we aimed to test if stimulation of lysosomal function selectively removes altered mtDNA or whether the beneficial effects observed are more related to bulk mitophagy

activation. We have previously shown that expression of K320E in skeletal muscle leads to the accumulation of mtDNA alterations, unfortunately only in very old animals, making this model not very convenient[23] (Fig. 1). However, expression of K320E in muscle satellite cells (Pax7-Cre[ERT]; K320E[msc]), followed by muscle damage induced by cardiotoxin, shows a rapid accumulation of mtDNA alterations only after one week of regeneration, leading to newly generated

**Fig. 6 | VPS35 coordinates mtDNA removal upon mtDNA damage.**
**a** Transmission Electron Microscope images of Twinkle-APEX2 cells contrasted with DAB to detect mitochondrial nucleoids. Arrows indicate mitochondrial membrane remodeling. **b** Electron tomography images and 3D reconstruction in cells expressing K320E-APEX2. 3D reconstruction was performed following membranes in 30–50 slices of a tomogram (distance of slices: 1.108 nm, $n = 5$). MIM, green; MOM, magenta; non mitochondrial membrane, blue. **c** Hierarchical clustering of Z-score-normalized protein targets enrichment of proteins related to vesicle trafficking, endosomes and lysosomes for Twinkle and K320E. Targets were selected based on significant enrichments compared to mitoAPEX2 (significance: q-value <0.05 and absolute log2 difference >1). **d** Hierarchical clustering of Z-score-normalized protein targets enrichment of proteins related to endosomes and lysosomes for K320E. **e** C2C12 cells expressing K320E-mCherry and labeled with α-RAB5 and α-VPS35 antibodies. Arrows indicate colocalization points. **f** Quantification of VPS35 particles in contact RAB5 ($n = 3$, >14 cells per replicate). **g** C2C12 cells expressing Twinkle-mCherry variants labeled with α-VPS35 and α-TOM20 antibodies and grown in basal medium or treated for 5 days with 50 ng/ml EtBr. Arrows indicate colocalization. **h, i** Quantification of VPS35 particles in contact with Twinkle ($n = 3$, >18 cells per replicate). Scale bar, 1 μm (**a**), 100 nm (**b**) or 10 μm (**e, g**). $P$ values calculated using paired two-tailed Student's $t$-test (**f**), or One-way ANOVA with Tukey correction for multiple comparison (**h** and **i**). Data is presented as Mean ± SEM.

cytochrome-c-oxidase (COX) negative fibers (stained blue, Supplementary Fig. 12a, b), while mtDNA copy number remained stable (Supplementary Fig. 12c).

Thus, we used this muscle regeneration paradigm in combination with rapamycin treatment to test if rapamycin can purify mtDNA alterations also in mammals in vivo. Regenerated muscles from vehicle-treated K320E$^{msc}$ mice showed a prominent accumulation of COX-negative fibers indicating mitochondrial dysfunction (Fig. 8a). In contrast, mice treated with rapamycin showed much less COX-deficient cells in the regenerated area (Fig. 8a–c). Consistently, the qPCR analysis revealed that mtDNA copy number remained unchanged (Fig. 8d), while mtDNA alterations were absent (Fig. 8e). We noticed that COX staining was much lighter in rapamycin-treated animals, suggesting a change in mitochondrial OXPHOS activity. Vehicle-treated mice showed a predominant accumulation of glycolytic fiber-type 2B in both WT and K320E$^{msc}$ mice (Supplementary Fig. 12d), while in rapamycin-treated WT mice, fiber-type staining showed the largest shift towards mitochondria-rich type I fibers (Supplementary Fig. 12e). Interestingly, regenerated fibers in K320E$^{msc}$ mice showed a mixed myosin heavy chain pattern after one week of regeneration. These data indicate that rapamycin can be used as a modulator of mtDNA turnover, which specifically eliminates mutated mtDNA species, thus ameliorating mitochondrial dysfunction, albeit with metabolic changes affecting the muscle fiber-type composition.

## Discussion

Autophagy and specifically mitophagy and its variants are well-established pathways for mitochondrial turnover, essential to maintain mitochondrial fitness[3]. Loss of mitochondrial quality control mechanisms, either by specific mutations of key players or by reduced autophagic activity, strikingly correlates with the acquirement of mtDNA mutations[51]. However, an exacerbated activation of mitophagy may affect cellular energy supply, leading to deleterious effects by reduction of the mitochondrial pool[52]. Therefore, the fine-tuned regulation of mitochondrial quality control mechanisms is crucial to maintain cellular energy homeostasis. Nevertheless, mitophagy, understood as the specific removal of the entire damaged organelle, does not provide the required selectivity to remove only mutated mtDNA. Hence, the existence of a specific turnover mechanism has been postulated, but till now, not yet proven[53].

Mutations in genes encoding for proteins involved in mtDNA replication, like the helicase Twinkle, and in mtDNA maintenance, lead to mitochondrial diseases, with brain and skeletal muscle being regularly affected. In addition, somatic mutations in mtDNA accumulate in many organs during normal ageing[2], leading to a tissue mosaic where few cells with mitochondrial dysfunction, caused by high mutation loads, are embedded in normal tissue[54]. Although toxic substances can cause mitochondrial damage, the most prevalent reason for mitochondrial dysfunction in healthy humans is the accumulation of alterations in mtDNA due to replication errors. In general, tissues most dependent on mitochondrial function are most severely affected when carrying mtDNA mutations. Paradoxically, we found that expression of the dominant negative K320E mutation of Twinkle in extraocular

muscle shows differential vulnerability of muscle fiber types, with mitochondrial dysfunction especially affecting fibers with a glycolytic metabolism[23]. In agreement with these results, we found less mtDNA alterations in aged SOL, a muscle rich in type I fibers, which mostly rely on mitochondrial ATP production, compared to the TA mostly composed of fast-twitch, glycolytic fibers. Noteworthy, different muscles rich in oxidative vs. glycolytic fibers show notable differences in the expression of genes involved in mitochondrial dynamics[55], making oxidative muscles more resistant to ageing related dysfunction[56]. In fact, our data shows that SOL expressing K320E already has an increased turnover rate, however this was not related to increased mitochondrial turnover. Nevertheless, the activation of this mechanism was not enough to counteract the accumulation of mtDNA damage, at least in aged mice.

In contrast to terminally differentiated muscle of aged mice, proliferating cells in culture did not accumulate mtDNA alterations upon expression of K320E. Our data demonstrates that interfering with mtDNA replication, either in cells carrying the mutated form of the mitochondrial helicase or mild treatment with EtBr, induces the accumulation of oxidative damage and increases the relative number of lesions in the mtDNA. We demonstrate that mtDNA damaged-induced depletion is caused by a specifically stimulated mtDNA turnover, which is ATG5 dependent and requires lysosomal function. Due to the multiple roles found for ATG5, such as membrane binding and curvature or lysosomal maturation, its specific role here remains to be elucidated[57,58].

To determine the proteins involved in mtDNA turnover upon mtDNA damage, we used spatial proteomics using the ascorbate peroxidase APEX2. Proximity ligation enables restricted labeling of neighboring proteins and allows their identification by MS, allowing the determination of even transient or weak protein interactions, which can escape from traditional pull-down methods[43]. Thus, we have identified that mitochondrial nucleoids carrying mtDNA damage localize preferentially in specific mitochondrial regions characterized by cristae remodeling. The area of influence of such nucleoids is enriched with proteins related to pore formation (VDAC1, VDAC2, VDAC3, TSPO), protein import (TOM40, SAMM50), solute carriers (SLC25A11, SLC25A20, SLC25A1, SLC25A4, MPC2, ABCD3, SFXN1) and inner membrane scaffolding proteins (IMMT/MIC60, CHCHD3, CHCHD6). The identification of proteins located in a different sub-mitochondrial compartment than the mitochondrial nucleoid, evidences the membrane readjustments necessary to extract mutated nucleoids and shows how mitochondrial inner and outer membrane might be modified in order to expose mitochondrial matrix components to the outer membrane barrier. Indeed, we observe specific changes in mitochondrial cristae structure, local changes in membrane potential, and importantly, differential protein interaction patterns upon mtDNA damage.

Three proteins present in three cellular compartments are responsible for mtDNA distribution and selective turnover: ATAD3 in the mitochondrial inner membrane, SAMM50 in the mitochondrial outer membrane and, VPS35 in the cytoplasm. Twinkle arises as the link between nucleoids and the inner membrane through interaction

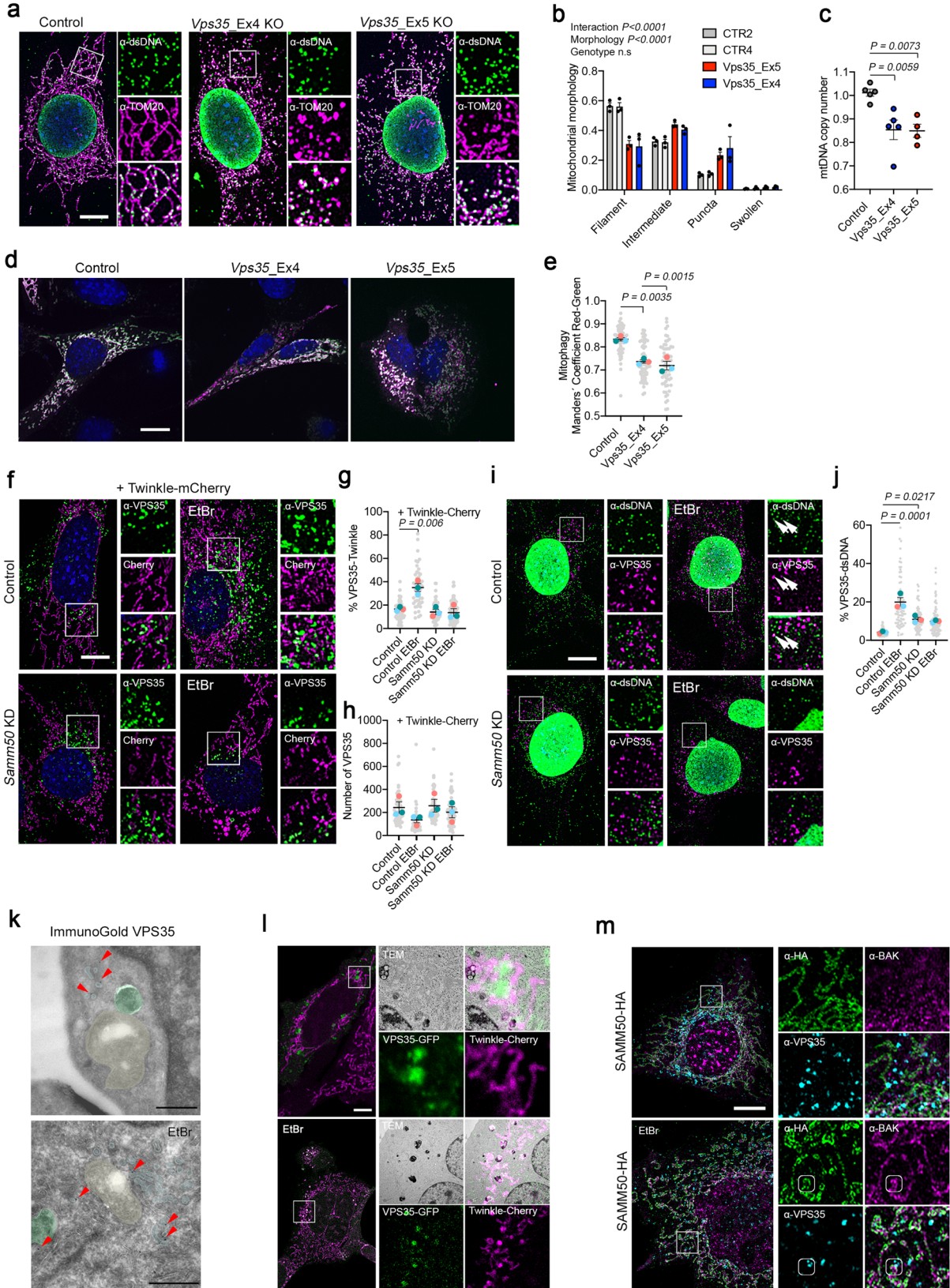

with ATAD3. Interestingly, the human orthologs ATAD3A and ATAD3B participate in membrane dynamics and quality control. ATAD3A has a pivotal role in the organization of the inner mitochondrial membrane and nucleoid distribution in cooperation with the MICOS complex[59], while ATAD3B is a mitophagy receptor for mtDNA damage induced by oxidative stress[60]. SAMM50, which resides in the mitochondrial outer membrane, also interacts with the MICOS complex and controls, among several other functions, the mitochondrial membrane architecture[61]. Interestingly, both ATAD3 and SAMM50 interact with the mitochondrial fission factor Drp1[62,63] and regulate PINK1-Parkin[47,64], thus providing the link between nucleoid localization and specific degradation of mtDNA.

**Fig. 7 | SAMM50 and VPS35 are required for nucleoid and specific mtDNA elimination. a** α-TOM20 and α-dsDNA immunofluorescence of *Vps35* KO MEFs and **b** Mitochondrial morphology quantification (*n* = 3, >30 cells per replicate). **c** mtDNA copy number of *Vps35* KO cells (*n* = 5 independent cultures). **d** Control and *Vps35* KO cells transfected with Fis1p-GFP-mCherry plasmid to detect canonical mitophagy. Red signal represents mito-lysosomes. **e** Manders' coefficient quantification of transfected cells. A decrease in Manders' coefficient indicates canonical mitophagy activation (*n* = 3, >20 cells per replicate). **f** α-VPS35 labeling in control and *Samm50* KD cells transduced with Twinkle-mCherry and treated with EtBr. **g** Quantification of VPS35 contact site with Twinkle and **h** number of VPS35 foci per cell (*n* = 3, >15 cells per replicate). **i** α-VPS35 and α-dsDNA immunofluorescence in control and *Samm50* KD cells. **j** Quantification of dsDNA foci in contact with VPS35 endosomes (*n* = 3, >20 cells per replicate). **k** Immunogold labeling of VPS35 in steady state and EtBr treated cells. Gold particles are signalized by red arrows. Mitochondria were colored in yellow, endosomes in cyan and late autophagy organelles in green. **l** Correlative Light-Electron microscopy in cells expressing Twinkle-Cherry and transiently transfected with VPS35-GFP, in steady state and after EtBr treatment. (*n* = 5 transfected cells). **m** Representative images of Airy Scan Super-Resolution microscopy of cells transduced with SAMM50-HA and labeled with α-HA, α-VPS35 and α-BAK, in steady state and upon EtBr treatment. Small frames mark colocalization of three proteins (marked in three channels). Scale bar, 10 μm (**a**, **d**, **f**, **i**, **l**, and **m**) and 500 nm (**k**). *P* values calculated using One-way ANOVA with Tukey correction for multiple comparison (**c**, **e**, **g**, **h**, and **j**) and Two-way ANOVA for Genotype and Morphology interaction (n.s = no significative). Data is presented as mean ± SEM.

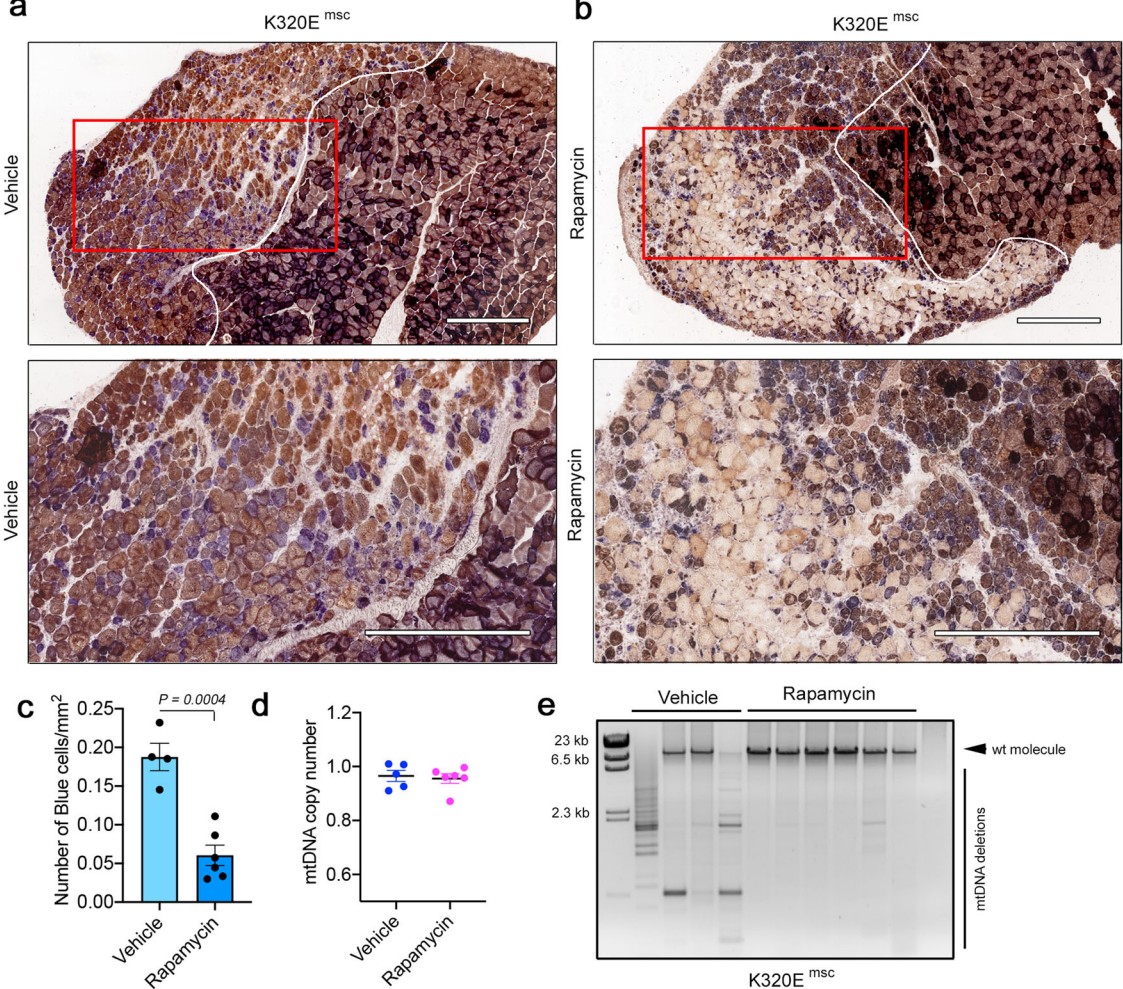

**Fig. 8 | Rapamycin eliminates mtDNA deletions without affecting copy number in vivo.** COX-SDH staining of regenerated TA muscle from Pax7-K320E mice (K320E^msc). After cardiotoxin-induced injury, mice were injected for 5 days either with **a** vehicle or **b** 2 mg/kg Rapamycin. **c** Quantification of COX-negative cells (blue) in the injured area. **d** mtDNA quantification by qPCR or **e** Long-range PCR in regenerated muscle from K320E^msc mice treated with vehicle or with Rapamycin. (Vehicle, *n* = 4; Rapamycin, *n* = 6). Scale bar, 500 μm. *P* values were calculated using the Unpaired Student's *t* test. Data are presented as Mean ± SEM.

Besides the classical functions of mitochondria in metabolism, Ca²⁺ handling, and apoptosis, mitochondria are now considered central hubs in regulating inflammatory responses[42]. Disruption of mitochondrial integrity triggers mtDNA release into the cytosol, where it can elicit the innate immune response either through the expression of type I interferons or via the inflammasome, leading to the activation of genes for pro-inflammatory cytokines. Here, and in agreement with recent research[37], we demonstrate that SAMM50 controls BAK distribution in the mitochondrial outer membrane and *Samm50* KD leads to mtDNA release and activation of the innate immune response. Remarkably, BAK/BAX macropores control mitochondrial herniation

and inner membrane permeabilization[39], which might facilitate mitochondrial nucleoids to trespass the mitochondrial inner barrier. In addition, lack of SAMM50 also induces mitophagy, but excluding mtDNA for degradation[47]. All these data together suggest that SAMM50 confers the required selectivity for mtDNA removal, controlling the activity of BAK. Whether this pathway can be modulated to avoid the activation of the inflammatory response following mtDNA damage remains to be elucidated.

Our data underline the role of endosomes in this quality control pathway for mtDNA. Electron tomography, followed by image reconstitution of mitochondria with mtDNA damage, evidences the

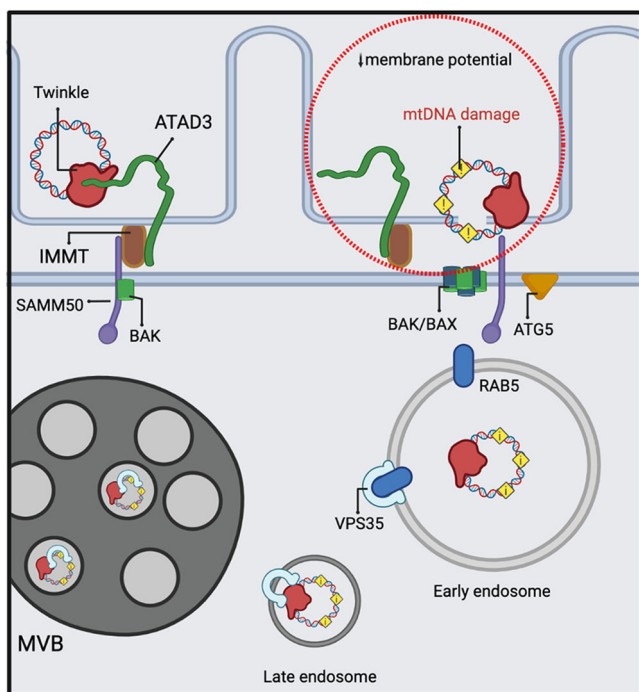

**Fig. 9 | Proposed model for mitochondrial nucleoid extraction upon mtDNA damage.** MtDNA damage induces local changes in membrane potential and cristae remodeling. ATAD3 and SAMM50 coordinate to extract mitochondrial nucleoids probably through BAK/BAX pores. Mitochondrial nucleoids containing mtDNA are engulfed or transferred to early endosomes. VPS35 mediates the maturation of these vesicles to late autophagy structures. The image was generated with full licensed BioRender.com.

presence of intricate membranes in close proximity to mitochondria, resembling organelles from the endo-lysosomal vesicle system. Using spatial proteomics, we were able to identify proteins, which have been related to unconventional forms of mitophagy. For instance, we detected the endosomal protein RAB5, linked to non-canonical mitophagy[44], and the RAB7 effector FYCO1, which mediates the binding of endosomes to LC3 and PI3P and allows maturation and lysosomal degradation[65]. We also show that VPS35 associates with mitochondria upon mtDNA damage, and nucleoid and dsDNA-containing particles are then directed to the lysosomal compartment. VPS35 recruitment is SAMM50 dependent and, upon mtDNA damage, such recruitment is specific to nucleoid-containing regions. SAMM50 is required for the specific transfer of nucleoids to VPS35 endosomes, which in a later step will fuse with late autophagy structures.

Quality control mechanisms of mitochondrial sub-compartments have been described to take place through MDVs[66]. Canonical MDVs are generated on the mitochondria in an ATG5 and LC3-independent manner, but our data support the essential role of ATG5 in mtDNA turnover. In addition, SAMM50 has been linked to piecemeal mitophagy, independent of MDVs, through direct interaction with the p62 adaptor[41]. Furthermore, VPS35 foci observed by immunogold staining and CLEM were always mitochondria independent or close to the endosomal compartment. Hence, these data support a model where SAMM50 serves as a platform to recruit components of the mitochondrial fission machinery, reorganizes membrane architecture, facilitates endosomal recruitment, and controls the elimination of damaged mitochondrial nucleoids (Fig. 9). However, due to the variability of the MDV machinery, which changes depending on the cargo, vesicle fate and the triggering insult, we cannot exclude that mtDNA turnover shares some MDV components, for example, for the formation of the mitochondrial extrusion containing mtDNA.

Nonetheless, we conclude that mtDNA removal upon mtDNA damage exemplifies a specialized non-canonical mitophagy trail involving endosomes.

Our data provide evidence that modulation of autophagy can be used as a strategy against the accumulation of mtDNA mutations in vivo. Lysosomal biogenesis is regulated by the serine-threonine protein kinase mTORC1, which resides on the lysosomal surface. mtDNA replication defects activate mTORC1 and the integrated mitochondrial stress response in a cascade of effects, with wide downstream consequences[67]. It is well-known that mTORC1 activation inhibits autophagy by influencing both the formation of autophagosomes but also endosomal acidification and lysosome formation[68]. In order to prove that stimulation of mTORC1 can be used for clearance of mtDNA-bearing mutations in vivo, we used a mouse model where mtDNA alterations rapidly accumulate. Indeed, activation of the mTORC1 pathway by rapamycin was able to eliminate abnormal mtDNA molecules without affecting mtDNA copy number, and thus reduce the accumulation of cells with mitochondrial dysfunction. Rapamycin has been described as a potential treatment against mitochondrial diseases[20], however, in muscle, mTORC1 inhibition also provokes a fiber-type shift. Therefore, further experiments directed to discover specific agonist inducing mitochondria-endosomal recruitment, while avoiding activation of undesirable effects, are needed.

VPS35, which has been extensively linked to neurological diseases such as Parkinson's and Alzheimer's disease, appears as a regulator of mtDNA quality control necessary to maintain mitochondrial intactness. Modulation of *VPS35* expression has been evaluated as a potential approach against Parkinson's disease[69] and, in *Drosophila*, *Vps35* overexpression can rescue an LRRK2-induced Parkinson's phenotype[70]. Interestingly, iPSC-derived neurons from *LRRK2*-PD patients showed an accumulation of mtDNA damage[71]. The ability of VPS35 to eliminate mtDNA-bearing mutations in human disease-related models needs to be further explored as a potential therapeutic strategy.

In conclusion, we unveil a complex mechanism with physiological relevance for mitochondrial fitness. Upon mtDNA damage, mitochondrial nucleoids are eliminated through an endosomal-mitophagy-related pathway. Twinkle mediates nucleoid binding to the mitochondrial inner membrane through ATAD3 interaction, which is responsible for nucleoid organization. SAMM50 provides the required specificity to eliminate mtDNA while VPS35 supplies the selectivity. Interestingly, mutations in *TWNK*, *ATAD3A*, and *VPS35* are linked to several severe mitochondrial diseases having in common mtDNA instability[72–74], therefore representing a cluster of proteins involved in specific mtDNA turnover.

## Methods

### In vivo experimental approaches

Mouse lines were generated in C57BL6J congenic and maintained in a pure background. In all experiments, both male and female mice were used. As controls Knock-in littermates from the same strain without the Cre allele were used. Mice were maintained in individually ventilated cages, maintained at 23 °C, 12:12 h light–dark cycle, with specified pathogen-free hygiene levels, free access to water and a regular chow diet ad libitum (Sniff V1554-300), and monitored regularly for signs of suffering. K320E transgenic (point mutation K320E; Rosa26-Stop-construct; downstream EGFP) mice were generated previously by our group[24]. Mice expressing Cre recombinase under the control of the skeletal muscle-specific MLC1f- promoter or satellite cells Pax7-Cre[ERT] were generated by crossing R26-K320EloxP/+ mice with mice expressing Cre recombinase under the control of the skeletal muscle-specific MLC1f- promoter or satellite cells Pax7-Cre[ERT].

All procedures and experimentation with mice were performed according to protocols approved by the local authority (LANUV, Landesamt für Natur, Umwelt und Verbraucherschutz NRW, approval

number: 2019-A090). Autophagic flux was tested by intraperitoneal injection of 50 mg/kg chloroquine 4 h prior to euthanasia. Activation of Pax7-Cre[ERT] promotor was performed by daily intraperitoneal injection for 5 days, of 2 mg tamoxifen dissolved in miglyol. For muscle regeneration experiments, 2 days after the last tamoxifen injection, mice were anesthetized with 2%Xylazin, 10% Ketamine in NaCl 0.9% and 10 μM Cardiotoxin (*Naja Pallida*, Latoxan) was injected inside the TA fascia. After 2 days of rest, 2 mg/kg rapamycin dissolved in miglyol was injected intraperitoneally, daily for 5 days.

### Generation and culture of cell lines
Information regarding vector generation can be found in the Supplementary information file. C2C12 cell line was purchased from ATTC and grown in DMEM 4.5 g/L Glucose + GlutaMax, 1× Pen/Strep (complete DMEM) supplemented with 20% FBS. Immortalized mouse embryonic fibroblast *Atg5* WT and *Atg5* KO MEFs and immortalized MEFs line were maintained in complete DMEM containing 10% FBS. Stable cell lines were generated by transducing C2C12 cells or MEFs with pBabe-Puro retroviruses. Briefly, HEK293 cells were plated in 10 cm² dish transfected with pCL-ECO and pBABE-Puro vectors using PEI. After 48 h and 72 h the medium, containing viruses, was harvested, filtered through 45 μm, mixed with Polybrene (final concentration 10 μg/ml), and added to target cells. 48 h post transduction, positive clones were selected by adding Puromycin at 2.5 μg/ml to the medium in C2C12 and 5 μg/ml to MEFs and maintained during all the experiments. shRNA clones were generated by transducing MEFs with pMKO.1-GFP vectors containing specific shRNA sequences (Supplementary Table 1). Prior to all experiments, the transduction rate was verified to be higher than 99% of GFP-expressing cells by Flow cytometry.

For a generation of Vps35 CRISPR Cas9 KO clones, MEFs were transiently transfected with the vector containing gRNA and selected with 3 μg/ml puromycin for 4 days. Single clones were plated independently using cloning cylinders and analyzed by western blot. Total DNA was isolated from VPS35-negative cells and genomic DNA modification was verified by Sanger sequencing. Exon 4 and Exon 5 were amplified (Supplementary Table 2) and cloned using pJET1.2 cloning kit (Thermo Fisher) before sequencing.

### Transfection and chemical treatments
In the corresponding experiments, transient transfection was achieved using Lipofectamine 3000 following manufacturer instructions. Plasmids from other sources independent of us but used in this work for transient transfection were: LC3-GFP (Addgene #21073), LAMP1-GFP (Addgene #34831), LC3-GFP-mCherry (kindly provided by Dr. Terje Johansen), Fis1p-GFP-mCherry and MAPL-GFP from CECAD Imaging Facility.

To assess mtDNA damage, cells were treated with 200 μM H₂O₂, 30 μM CCCP for 4 h or 72 h with media containing 50 ng/ml EtBr and 50 μg/ml Uridine. Different steps of autophagy were blocked using 5 mM 3MA, 20 μM SBI0206965, or 10 μM Chloroquine for 24 h.

### mtDNA amplification and qPCR
Total DNA was isolated using DNeasy Blood & Tissue Kit (Qiagen) according to the manufacturer's instruction. 25 ng of total DNA was used for the analysis of threshold amplification differences between mtDNA and nuclear DNA (delta C(t) method with specific primers (Supplementary Table 2). Long-range PCR was used to screen for the presence of mtDNA alterations. 14 Kb of mtDNA was amplified using Rabbit Bioscience Long Range kit with oligos described in Supplementary Table 2.

Quantitative real-time PCR was performed using cDNA retrotranscribed from 1 μg RNA using PowerUp SYBR green (Thermo Scientific) and corresponding oligos specified in Supplementary Table 2. Quantitative analysis of mtDNA damage was performed using long-run

rtPCR method[34]. Briefly, a concentration curve was performed for each sample using short mtDNA oligonucleotides (117 b.p amplicon; Supplementary Table 2) and PowerUp SYBR green, followed by a long-run rtPCR with long mtDNA oligonucleotides (10 Kb amplicon; Supplementary Table 2), SYTO-9 as a fluorescence reporter and Platinum Pfx Polymerase (Thermo Scientific). Before the experiment, SYTO-9 concentration was determined and the instrument calibrated. All experiments were performed on Quant Studio 1 qPCR (Thermo Scientific) with at 2–3 technical replicates and with at least three independent samples. mtDNA damage was quantified by normalizing the concentration value of the sample to the internal reference (short amplicon). Values are converted to relative lesion frequencies per 10 Kb by applying the Poisson distribution (lesions/amplicon = −ln (A$_t$/A$_0$), where A$_t$ is the normalized concentration value for treated samples and A$_0$ for the control.

### Histology, immunofluorescence and microscopy
For tissue histology, mice were sacrificed by cervical dislocation, muscles dissected, mounted in cork with OCT medium (Tissue-Tek), snap-frozen in isopentane cooled in liquid nitrogen, and stored at −80 °C until needed. 10 μm thick sections covering the injured area were produced using a cryostat maintained at −20 °C (Leica CM 3050 s, Techno-med). To assess the integrity of mitochondrial function, the sections were sequentially stained for COX and SDH activities. Frozen sections were incubated 20 mins at 37 °C in COX solution (20 mg/ml catalase, 74 mg/ml sucrose, 2 mg/ml cytochrome c, and 1 mg/ml DAB in 50 mM Na₂HPO₄ pH 7.4). After PBS wash, sections were incubated for 30 min at 37 °C in SDH staining solution (2 mg/ml NBT, 0.2 M Sodium succinate, 50 mM MgCl₂, 50 mM Tris-HCl, pH 7.4), washed 3 times with Milli-Q water, and mounted in Glycerol gelatine medium (Sigma).

For tissue immunofluorescence, mice were perfused with 4% PFA in PBS prior to muscle collection. Samples were equilibrated in 15% sucrose for 6 h and 30% sucrose overnight before being frozen in OCT medium. For LC3 IF, samples were pre-incubated with 0.1% SDS for 5 min. Antibody specificity was determined in muscle sections from LC3-GFP transgenic mice (kindly provided by Dr. Evangelos Kondilis). Cryosections were blocked for 1 h with 1% Western blocking reagent (Roche) containing 0.1% Triton in PBST, antibodies incubated overnight at 4 °C and secondary antibodies at room temperature for 1 h in blocking buffer. Samples were mounted in Fluoromount G (Thermo Fischer) containing DAPI. Fiber-type staining was performed as described previously[29]. Images were obtained with Leica SP8 with 63x/1.40 oil PL Apo-objective.

For in vitro immunofluorescence, cells were fixed in 4% PFA/PBS, permeabilized with PBS-0.2% Triton-X1000 for 30 min, and blocked for 1 h at RT in blocking buffer (5% fat-free milk powder, 10% FBS, 1% BSA, 0.1% Triton-X-100 in PBS). Primary antibodies were incubated in a blocking buffer overnight at 4 °C and secondary antibodies for 1 hour at RT. mtDNA replication rate was determined by pulse BrdU labeling. Briefly, cells were incubated with 20 μM BrdU (Sigma) for 4–6 h, fixed with 4% PFA/PBS for 30 minutes, and directly permeabilized with 0.5% Triton X-100 on ice for 5 min. To allow access to mtDNA, cells were incubated with HCl 2 N for another 30 min prior to immunofluorescence.

Antibodies used for immunofluorescence were: rabbit polyclonal α-V5 (1:500) (Thermo), rabbit polyclonal α-TOM20 (1:500), mouse monoclonal α-VPS35 (1:500), α−8-OHdG (1:250), α-V5 (1:100) (Santa Cruz); goat polyclonal α-VPS35 (1:500), mouse monoclonal α-dsDNA (1:1000) and rabbit polyclonal α-RAB5 (1:500) (Abcam); rabbit polyclonal α-LC3 (1:200), α-p62 (1:200), α-LRPPRC (1:400), α-TOM20 (1:1000), α-BAK (1:500), (Proteintech); monoclonal α-BrdU (1:1000) (BD Bioscience); monoclonal α-HA (1:500) (Sigma) and polyclonal α-HA (1:500) (Cell Signaling). For fiber type identification the following primary antibodies were used: α-MyHC-I (1:100) (BA-D5), α-MyHC-2A

(1:100) (SC-71), and α-MyHC-2B (1:100) (BF-F3) (Developmental Studies Hybridoma Bank).

Fluorescence secondary antibodies (1:1000) goat α-mouse, α-rabbit Alexa Fluor-488, 555 and 647, rabbit α-goat-647 and donkey α-mouse-488, α-mouse-555, α-rabbit-647 and α-goat-Rhodamine were used accordingly to the primary antibodies. Additionally, α-mouse IgM Alexa Fluor-488, α-mouse IgG Alexa Fluor 555, and α-mouse IgG2b Alexa Fluor 647 were used for fiber-type triple staining.

Coverslips were mounted using DAPI-Fluoromount G. Images were acquired using a spinning-disk confocal microscope (Ultra View VoX; PerkinElmer) with a ×60/1.49 oil PL-Apoobjective (Nikon), Leica SP5 microscope controlled by LAS AF 3 with ×2.5 extra magnification, Leica SP8 with ×63/1.40 oil PL Apo-objective and Airy Scan Confocal Microscope (Zeiss) with ×63 Plan-Apochromat/1.4 Oil DIC.

## Mitochondrial cristae structure, membrane potential, and super-resolution microscopy

For mitochondrial membrane potential cells were grown for 72 h in a complete medium or EtBr-containing medium. The membrane potential was assessed by flow cytometry using TMRE (BD Pharmigen). Briefly, cells were trypsinized and resuspended at a concentration of $1 \times 10^6$ cells/ml in media containing 100 nM TMRE or 100 nM TMRE + 60 μM CCCP as control and incubated 30 min at 37 °C. Cells were washed twice in PBS containing 2% FBS and 2 mM EDTA, filtered using a 30 μm cell strainer, and measured using a BD FACS Canto II Flow Cytometer (BD Biosciences). At least 20,000 events from each sample were recorded. Data were analyzed using FlowJo 10.8.1 software (BD Biosciences). For the analysis of membrane potential in Airy Scan, cells were grown in glass bottom plates (Ibidi) and incubated for 1 h in medium containing 15 nM TMRE. Images were acquired with Airy Scan Confocal Microscope (Zeiss) with ×63 Plan-Apochromat/1.4 Oil DIC. For cristae architecture, cells were treated as described before and stained for 30 min with 250 nM PK Mito Orange dye (GenVivo Biotech) and imaged after 2 hours with TCP SP8 gSTED Leica Microsystems using PL Apo ×100/1.40 Oil STED Orange.

## Western blot and co-immunoprecipitation

Cells pellets were lysed with RIPA buffer (150 mM NaCl, 1% Triton-X1000, 0.1% SDS, 50 mM Tris-HCl pH 8, 0.5% Na-deoxycholate) containing protease inhibitor (Roche) and protein concentration measured using the Bradford assay.

For immunoprecipitation, cells expressing Twinkle-APEX2-V5, SAMM50-HA, ΔPOTRA-SAMM50-HA, ATAD3-HA, and ΔIMS-ATAD3-HA, were pelleted and solubilized in IP Buffer (500 mM HEPES KOH pH 7.2, 150 mM NaCl, 1 mM MgCl₂, 1% Triton-X1000 and Protease inhibitor (Roche). 500 μg of total protein extract was used to IP with 2.5 μg α-V5 rabbit polyclonal antibody, α-VPS35 mouse monoclonal or α-HA rabbit polyclonal or mouse monoclonal, depending on the experiment, overnight at 4 °C, and recovered after incubating for 6 h at 4 °C in a rotator with equilibrated Agarose Protein-G beads (Abcam). Samples were washed 5 times with washing buffer (10 mM HEPES KOH pH 7.2, 150 mM NaCl, 1 mM MgCl₂, 0.2% Triton-X1000) and once with PBS. For western blot analysis, washed agarose beads were resuspended in 2x Laemmli buffer and boiled for 10 min.

Proteins were transferred after electrophoresis to a PVDF membrane previously activated with methanol. Membranes were blocked (5% milk in TBS-0.1% Tween-20) and incubated overnight with primary antibodies. Antibodies used in this work are: monoclonal α-HA (1:1000) (Sigma), monoclonal α-V5 (1:1000) (Abcam), polyclonal α-V5 (1:1000) (Thermo Scientific), polyclonal α-TOM20 (1:1000) and monoclonal α-VPS35 (1:1000) (Santa Cruz), monoclonal (1:200) and polyclonal (1:1000) α-SAMM50 (Santa Cruz and Abcam respectively), polyclonal α-HA (1:1000), α-ATAD3 (1:1000), α-LC3 (1:1000), α-p62 (1:1000), α-TOM40 (1:1000), α-TOM20 (1:1000), α-MIC19 (1:1000), α-IMMT/MIC60 (1:1000) and α-LRPPRC (1:1000) (Proteintech),

polyclonal α-GAPDH (1:2000) (Novus Biologicals) and rodent OXPHOS cocktail (1:2000) (Abcam), polyclonal α-BAK (1:1000) (Cell Signaling). Secondary goat anti-mouse, goat anti-rabbit, and goat anti-chicken HRP (1:10000) (Jackson Laboratory) were used accordingly to primary antibodies. Images were acquired using the ECL Advanced Chemiluminescence kit (GE Healthcare Life Sciences®, UK) according to the manufacturer's protocols and visualized using a LAS500 CCD camera.

## APEX2 proximity biotinylation

Cells transduced with retroviral vectors containing Twinkle-APEX2-V5, K320E-APEX2-V5, and mitochondrial matrix targeted APEX2-V5 (mitoAPEX2), were used for proximity biotinylation experiments. Biotin-phenol labeling was performed by incubating transgenic C2C12 with 2.5 mM biotin-phenol for 7 h. For crosslink, 1 mM H₂O₂ was added to the cells and incubated for 1 min at room temperature. Quenching the reaction was achieved by washing 4 times with 1 mM sodium azide, 1 mM sodium ascorbate, and 5 mM Trolox in PBS. The cells were recovered and solubilized with RIPA Buffer containing 10 mM sodium ascorbate, 10 mM sodium azide, 5 mM Trolox, and Protease Inhibitor. Prior to an analysis by MS, biotinylation reaction was verified by SDS-PAGE using Streptavidin-HRP (1:2000) (Merck).

Streptavidin-magnetic beads (New England Biotech) were used to purify APEX2-induced crosslinked proteins. In such cases, 500 μg of total protein extracts containing biotinylated proteins were used and prepared for MS analysis. Samples were washed three times with RIPA buffer followed by three more washes with ABC buffer. Samples were denatured with 50 μl of urea buffer (6 M urea, 2 M thiourea) and followed by disulfide-bridge reduction using dithiothreitol at a final concentration of 5 mM for 1 hour at room temperature. To alkylate oxidized cysteines, 2-Iodoacetamide was added to the samples until a concentration of 40 mM was reached and incubated for 30 min in the dark. Lys-C was added in a ratio of 1:100 (0.1 μg enzyme for 10 μg protein) and incubated for 2–3 hours. Samples were finally diluted with ABC buffer to reach 2 M urea concentration. Protein digestion was performed overnight with trypsin 1:100. Samples were acidified with 1% formic acid and desalted using a modified version of the Stop and Go extraction tip (StageTip) protocol.

Alternatively, for immunofluorescence of biotinylated proteins, cells were seeded onto glass coverslips and after crosslink and quenching, fixed with 4% PFA containing 1 mM Sodium Azide, 1 mM Sodium Ascorbate and 5 mM Trolox. Biotinylated proteins were detected using α-Neutravidin-488.

## Pulsed SILAC labeling in mice and in-solution digestion

For pulsed SILAC labeling mice, 30–40 weeks old mice for K320E; Mlc1 line (C57BL/6 J) were fed a $^{13}C_6$-lysine (Lys6)-containing mouse diet (Silantes) for 14 days to monitor newly synthesized proteins by comparing the incorporation of Lys6 with the naturally occurring Lys-0. Mice were sacrificed at the end of day 14 and tissues dissected and snap-frozen in liquid nitrogen. Samples were grinded and proteins extracted and denatured by the addition of 4% SDS in PBS. To remove residual SDS, proteins were precipitated overnight in 4× ice-cold acetone (v:v). The following day, after centrifugation at 16,000 g for 10 min, the protein pellets were dissolved in urea buffer (6 M urea/2 M thiourea). The following protein digestion was performed as described previously but instead of overnight tryptic digestion, proteins were only digested with Lys-C (1:100 enzyme-to-protein ratio) for both pre-digestion (2 h at RT) and overnight digestion after dilution of urea using ammonium bicarbonate (ABC) buffer. Methods regarding MS analysis can be found in the Supplementary information file.

## Electron microscopy

All electron micrographs were taken with a JEM-2100 Plus Transmission Electron Microscope (JEOL) operating at 80 kV equipped with a OneView 4 K camera (Gatan). For CLEM, cells of interest were

identified using the obtained fluorescent and brightfield image. Reconstruction was done using ICY and ec-CLEM plugin.

For electron tomography, Ultrathin sections of 200 nm were cut using an ultramicrotome (Leica, UC7) and incubated with 10 nm protein A gold (CMC, Utrecht) diluted 1:25 in ddH20. Sections were stained with 2% Uranyl acetate for 20 min and Reynolds lead citrate solution for 3 min. Images and Tilt series for 1.108 nm thickness were acquired from −65° to 65° with 1° increment on a JEM-2100 Plus Transmission Electron Microscope (JEOL) operating at 200 kV equipped with a OneView 4 K 32 bit (Gatan) using SerialEM (Mastronarde, 2005). Reconstruction was done using Imod and Imaris. Specific information regarding sample processing for electron microscopy can be found in the Supplementary information file.

### Image analysis

All image analysis was performed on FIJI (NIH, Bethesda). LC3 and p62 puncta quantification were performed with the "counting cells" internal plugin for particles bigger than 2 pixels to exclude background. Fluorescence profile for cristae morphology and membrane potential was obtained with the plugin RGB Profiler. mtDNA foci quantification, VPS35-Twinkle analysis, VPS35-RAB5, VPS35-DNA, VPS35-SAMM50, TOM20-DNA, cytosolic dsDNA, 8-OHdG and BrdU analysis were performed with a self-created macro. Briefly, threshold was set for the different channels. Nuclear signal was selected and removed for the analysis. The signal corresponding to the mitochondrial network was selected and only the particles from the other channel bigger than 1 pixel inside or outside the mitochondrial network or in contact with the other channel, were considered for the analysis. Manders' coefficient was obtained using JaCOP plugin.

### Statistical analysis and reproducibility

For comparative analysis, $P$ values were calculated using specific tests indicated in figure legends. When significance was achieved, exact $P$ value is indicated in the figure. Analysis and Graphs were generated with Graph Pad Prism.

### Reporting summary

Further information on research design is available in the Nature Research Reporting Summary linked to this article.

## Data availability

The mass spectrometry proteomics data have been deposited to the ProteomeXchange Consortium (http://proteomecentral. proteomexchange.org) via the PRIDE partner repository[75] under the accession PXD023939. The data generated in this study are provided in the Supplementary Information/Source Data file, including uncropped versions of western blots. Source data are provided with this paper.

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

## Acknowledgements

We are thankful to CECAD Imaging and Proteomics facility for excellent technical support, especially Beatrix Martiny for immunogold prepara-tion. We thank Nadine Niehoff and Katrin Lanz for technical assistance, and Thomas Paß for scientific discussion. This work was supported by grants from the Deutsche Forschungs-gemeinschaft (PL 895/1-1) to D.P.M. and R.J.W. and Köln Fortune (341/2019) to D.P.M.

## Author contributions

Funding acquisition, D.P.M. and R.J.W.; conceptualization, D.P.M. and R.J.W.; investigation and formal analysis, D.P.M., A.S., S.K., F.G., K.M., J.B., J.H.; resources J.N., C.J., A.C.S., M.K.; analysis of MS data, S.K.; visualization, D.P.M.; writing-review & edit; D.P.M., R.J.W., S.K.; Super-vision, D.P.M. and R.J.W.

## Funding

## Competing interests

The authors declare no competing interests.
