## [Peer Review File · Nature Communications]

Mitochondrial membrane proteins and VPS35 orchestrate selective removal of mtDNA.Reviewers' comments:

Reviewer #1 (Remarks to the Author):

In this manuscript titled "Mitochondrial membrane proteins and VPS35 orchestrate selective removal of mtDNA", Pla-Martin et al. reveal that VPS35 and some mitochondrial proteins including Twinkle, ATAD3 and SAMM50 are involved in the remove of mutant mtDNA, and also showed that rapamycin eliminates mtDNA alterations in vivo. The findings shown in this manuscript are potentially interesting to the readers. However, the data in the manuscript are preliminary and do not strongly support some conclusions, and the mechanism of VPS35 and mitochondrial membrane proteins removing mtDNA is still obscure.

Major comments

1. How VPS35, Twinkle, ATAD3, and SAMM50 cooperate to remove mtDNA is still unclear, the pathway (mode) and the underlying mechanism are not clearly illustrated.

2. Although 8-OHdG assay is used in Fig. 2K, 2L, S4C, S4D, the assays detecting the damaged or mutant mtDNA are not enough, 8-OHdG assay and qRT-PCR assay should be performed throughout the manuscript to evaluate the integrity of mtDNA.

3. It is interesting that mtDNA instability induces the formation of mitochondrial protrusions, however, how mitochondrial protrusion is formed is not clear, and how mitochondrial protrusion delivers to lysosome is still unknown.

4. In Fig. 3, 5, S7, etc., the authors detected mtDNA removal by examining mtDNA copy number, how do the authors exclude the possibility that mtDNA biogenesis is decreased?

5. In Fig. 6, 7, S7, and S8, The authors used EtBr to damage mtDNA, some other mtDNA damage inducers such as H₂O₂, are required to further confirm the phenotypes.

Minor comments

1. Fig. 7g and 7i, the evidence of VPS35-Twinkle contacts is weak. The selected regions for quantification should not be the middle region where a lot of mitochondria exist, leading to being hard to be distinguished by confocal microscopy.

2. VPS35-mtDNA-Tom20 or VPS35-mtDNA-Twinkle immunostaining can be performed to evaluate VPS35-mtDNA contacts, and check the role of VPS35 in the formation of mitochondrial protrusions.

3. Line 264: Fig S5b should be Fig S6b.

Reviewer #2 (Remarks to the Author):

The manuscript by Pla-Martin and colleagues describes that mtDNA depletion after expressing a dominant negative version of the mitochondrial helicase Twinkle, or by treating cells with EtBr, is due to increased mtDNA turnover. Autophagy pathways were found to be important in this process but micro vesicles (MDVs) were not. The mitochondrial transmembrane proteins ATAD3 and SAMM50, participated in mitochondrial membrane remodeling required for mtDNA loss. The autophagy pathway also required the vesicular trafficking protein VPS35 which binds to Twinkle-enriched mitochondrial subcompartments upon mtDNA damage. Finally, using an in vivo muscle model, they showed that rapamycin selectively removes mtDNA deletions which accumulated during muscle regeneration, but without affecting mtDNA copy number. They concluded that mutant mtDNA elimination depends on these series of events and partners.

Overall, it is a very well-done study, where the authors put a great deal of thought and effort to figure out the fate of mutant mtDNA. Said that, I have a few concerns.

1) My main concern is the interpretation of EtBr-related experiments. As far as I remember, EtBr causes mtDNA level reduction by blocking mtDNA replication. mtDNA levels are reduce because cells continue to grow during treatment, reducing mtDNA levels by dilution. Although EtBr may induce low levels of mutations, the effect or prevalence of these would be extremely low compared to the effect in replication. Therefore, to equate loss of mtDNA during EtBr treatment to active elimination by autophagy is highly debatable. If, for example the ATAD3 KD cell line was growing well, the mtDNA levels should reduce. This brings up the cell growth of the cell lines when these analyses were conducted.

2) In Figure 1, the statement: "However, in SOL for K320E mice, chloroquine was not inducing a change in the LC3 ratio suggesting that, in this muscle, autophagy flux was already at maximum level in steady state." Is unclear. The LC3I and LC3II ratios do not appear to change in any sample, with or without chloroquine.

3) Regarding the use of LC3-GFP-mCherry or Fis1p-GFP-mCherry, the authors need to minimally have a sentence explaining that GFP is acid sensitive. Not every reader will be familiar with the system.

- 4) In Figure 8, the regenerated cells in the rapamycin treated look very abnormal, as not only cytochrome oxidase is absent, but also succinate dehydrogenase.
- 5) This comment (and ref 22) also raised some concerns: “Paradoxically, we found that expression of the dominant negative K320E mutation of Twinkle in extraocular muscle shows a remarkable differential vulnerability of muscle fiber types, with mitochondrial dysfunction especially affecting fibers with a glycolytic metabolism (22)”. The Cre used to delete the gene (the skeletal muscle-specific *Mlc1f*-promoter) is much more active in type II fibers.
- 6) Finally, I believe the final model is ok, but it is presented in a very strong/conclusive manner that is not warranted by (mostly) protein-protein interaction data. I would tone it down as “the data suggests a model”.

Reviewer #3 (Remarks to the Author):

In this manuscript by Pla-Martin et al, the authors report that mitochondrial membrane proteins in association with VPS35 removed damaged mitochondrial DNA through a selective autophagy process that require ATG5. The authors propose that the mitochondrial DNA helicase Twinkle direct the targeting of nucleoids to the autophagic pathway in collaboration with ATAD3 and SAMM50 to remove damaged mitochondria. They propose that this process involves vesicular transport protein VPS35 but is not dependent on the formation of mitochondrial derived vesicles. There is a lot of nice work presented in this manuscript, but they seem to support the already known pathways for the removal of mtDNA and mitochondrial fragments. Moreover, there are some conclusions which are not currently supported by the data provided here. I will first make two major points and then some minor points that need to be clarified.

1. The roles of mitochondrial membrane proteins

A): Twinkle and ATAD3: The new hypothesis proposed in this work is that Twinkle may control the targeting of mtDNA to the autophagic pathway. However, results showed so far do not support this conclusion. ATAD3B has already been reported to regulate the removal of damaged mitochondrial DNA (Shu et al, 2021; PMID: 33665835). Is Twinkle required for the removal of mtDNA? Results here do not support a role for Twinkle but only support the already known role of ATAD3 in clearance of mtDNA. To ascertain if Twinkle have any role(s) in mtDNA clearance, you will need to clarify if Twinkle interaction with ATAD3 is necessary and sufficient for mtDNA clearance or show that the process described here is not the same as the one already reported for ATAD3?

B): Twinkle and SAMM50. Does twinkle binds directly to SAMM50 and is this interaction relevant for clearance of mtDNA? SAMM50 is a beta barrel protein that is localized in the outer mitochondrial membrane. It is known to interact with the TOM complex and assemble both the MICOS and SAM

complexes. SAMM50 has also been reported to control basal removal of mitochondrial fragments without affecting the mitochondrial network (Abudu et al, 2021; PMID: 34037656). If Twinkle have roles in mtDNA removal through interaction with SAMM50, you will have to show how and the relevance of this interaction during this process. Is the SAM and MICOS complexes also involved? How do you separate the role of SAMM50 in basal piecemeal mitophagy from the clearance of mtDNA or are these the same process?

C). How does SAMM50 interact with VPS35 and is this relevant for this process. What about VDACS and TOM40 identified in VPS35 IP?

2. Roles of mitochondrial derived vesicles

A): The data provided here does not demonstrate whether mtDNA degradation occur through another pathway independent of MDV formation. VPS35-retromer complex is well known to regulate MDV formation and seem to be important in mtDNA degradation shown here. Is the mechanism of MDV formation by the VPS35 retromer complex different from the formation of these mtDNA containing fragments? Can we separate the role of VPS35 in MDV formation from the one you describe here? Are the components or cargoes of MDVs different from the components of the mtDNA fragments? Mutational analysis (on VPS35) is needed for the former while CLEM studies are required for the latter to properly differentiate these. The absence of MAPL or TOM20 or DRP1 is not sufficient to conclude that these fragments are not MDVs.

B): Another question is what does these fragments or particles contain and how are they shuttle to the lysosome or how are they recruited to autophagosomes? Electron tomography pictures (Figure 4) show that these fragments contain multi-membrane structures and not just nucleoids. Does these fragments contain inner mitochondrial membranes or proteins? Does it contain OMM or proteins? Are these fragments different from those degraded by SAMM50 mediated basal mitophagy (Abudu et al, 2021; PMID: 34037656) which degrades mitochondrial fragments independent of canonical mitophagy. Another important question is how to explain the importance of ATG5. Some specific MDV delivery to the lysosome is shown to be independent of LC3 and ATG5 (Soubannier et al., 2012; PMID: 22226745). The question here is how does these fragment fuse with ATG5-dependent autophagosome or how are they shuttle t the lysosome. Is the ATG8 proteins involved?

Minor Points

1. Expression of the K320E mutants seem to lead to altered mitochondria morphology (Fig S3) Can you quantify mitochondrial morphology.

2. It is now well known that OMM and IMM/Matrix proteins are degraded differently during mitochondrial quality control. Can you blot for IMM/Matrix protein as part of Fig. 1

3. Does expression of K320E drives basal mitophagy. Figure 2C is difficult to interpret as lack of mitophagy. Use Matrix targeted Keima or- tandem tag to show basal mitophagy levels.

4. Explain how you quantified nucleoids and describe in detail (preferably in materials and method) how you quantified mtDNA copy number. It is difficult to reconcile reduction in mtDNA copy number with no changes in nucleoids. Can you stain with mtDNA antibody and then do quantification in all these conditions?

5. You showed that K320E expression reduced the mtDNA copy number in both WT and ATG5 KO cells, but the nucleoid number do not change in ATG5 KO pointing to an ATG5-dependent removal of nucleoids. However, the copy number can be rescued in ATG5 WT with chloroquine treatment cells but not in KO cells. This statement is contradictory. This implies that mtDNA copy number is reduced in lysosomal-dependent manner but not ATG5 dependent manner, while nucleoids are removed in ATG5 dependent manner. Can you explain this.

6. You then suggested that Loss of ATG5 seems to affect mtDNA replication during K320E expression. Shouldn't mtDNA replication affects mtDNA copy number? If ATG5 KO (or autophagy) affects mtDNA replication, it should also affect mtDNA copy number. Can you explain this? Is this specific to ATG5 or is it autophagy per se? However, Ethidium bromide treatment leads to reduction of both nucleoids and mtDNA copy number in WT cells and not ATG5 KO cells. Why is this different from K320E expression? Are these two different processes?

7. Alternate to ATG5, can you use inhibitors to ULK complex or PI3K complex or another ATG KO (FIP200, ATG7 or ATG9) cells to corroborate the role of autophagy.

8. i. In Fig3a, why is the pBabe picture is also positive for V5 staining?

ii. Change Fig. S4c,e to S4e,f; line 193

iii. Check Graph labeling for Fig. 3L

iv. Change Fig3d to 4d; line 220

v. Change Fig S5b to S6b; line 264

vi. Change Fig. S8a,b to S9a,b; line 325

vii. Change Fig. S8c, to S9c; line 326

Reviewer #4 (Remarks to the Author):

Pla-Martin et al report on a new, autophagy-dependent recycling pathway for the removal of mutated/altered mtDNA, involving the mitochondrial DNA helicase Twinkle, the inner membrane protein ATAD3, the outer membrane protein SAMM50 and the vesicular trafficking protein VPS35. In their experiments, they make use of the disease-associated Twinkle variant K320E, which causes mtDNA alterations. They study the resulting cellular recycling responses in vivo (mouse muscle tissue) and in vitro (C2C12 cells and MEFs). Methodologically, they employ a wide variety of approaches, including (q)PCR, WB, Fluorescence microscopy, electron microscopy and proteomics.

I acknowledge the amount of work invested in this study and I believe that if rigorously done, the findings claimed here would be of value for the scientific community. However, I do not see this manuscript as a suitable candidate for Nature Communications due to the below detailed reasons / comments. In brief, the presented data does not (sufficiently) support the drawn conclusions. To my opinion, the manuscript does not convey information in a clear and coherent manner (inconsistencies across figures, suppl tables and main text). Due to my own area of research, I mainly focus on the presented proteomics data.

Proteomics data

pSILAC (Figures 1h, S2): State of the art pSILAC experiments, whether done in cell culture (see PMID: 29414762) or in mouse model (see PMID: 30315172), track label incorporation over several timepoints. Based on curve fits, these timepoints serve as an additional quality control layer and allow to (quite accurately) calculate protein turnover rates. These turnover rates can then be compared across samples and between proteins. Including a label-switch (chasing e.g. 2 Reps from L to H and 2 Reps from H to L) into the experimental layout further increases the reliability of the data. Here, the authors only used 1 timepoint, which delivered H/L ratios. In order to compare WT to KE across muscle samples, the authors calculated ratios of these H/L ratios, which amplifies inaccuracies in quantification. Also, a label-switch has not been done, although I do acknowledge that this is much easier done in cell culture than in mouse experiments. Finally, simply doing a Student's t-test is no longer state of the art. Instead, some kind of FDR-correction should be included (giving rise to adjusted p-values, termed q-value by MaxQuant / Perseus; see also below). By applying these improvements, the results would be much more reliable. The conclusions that can be drawn from the pSILAC data presented here are limited.

Twinkle pulldown (Figures 5a, 7e): For AP-MS experiment, the authors relied on label-free quantification using MaxQuant. As already mentioned for pSILAC, a standard t-test p-value is used instead of the adjusted q-value. However, the authors provide q-values in Table S2. When requiring a q-value of <0.05 for significance, which is more appropriate, ATAD3 is no longer significantly enriched and the only proteins that remain significant are Peo1, Rps29 and Dlat. In Figure 7e, no protein remains significant based on q-values. In lines 235-237, the authors write: "As expected, the majority of proteins interacting with Twinkle were related to mtDNA replication, transcription and translation, but only a few interactions were found to be significant". This claim is unproven, the authors should show this via a GO-Enrichment analysis. Nevertheless, an enrichment of non-significant proteins is not reliable. About the

ratio of enrichment (and ignoring statistics) for Atad3 and Samm50 (5a and 7e, respectively): With ~4-fold enrichment, Atad3 could be a genuine (weak) interactor, which is also shown in Figure 5b. However, in 5b, the authors should add a mitochondrial control protein that is not enriched in order to prove their point. For Samm50 (Figure 7e), with an almost 8-fold difference, the protein is of much higher abundance in the KE mutant. However, looking at Table S2, this is solely due to a very low abundance in the WT. When comparing KE to the Control sample, less than a 2-fold enrichment can be observed. Samm50 is more than 4-fold less abundant in the WT than in the empty vector control. How do the authors explain this observation? This also contradicts 5f. In 5f, the pulldown data generally looks quite weak. Taken together, to me, an Atad3 interaction seems possible/likely, albeit weak, for Samm50, the data is not convincing.

Vps35 pulldown (Figures 7a, 7b, S8b): Here, the authors appropriately use the q-value for statistics (why here and not before?). What is concerning, however, is that significant proteins are about as frequently depleted as they are enriched (see for instance 7b). This is somewhat atypical for an AP-MS experiment, because only interactors should significantly differ from to the control. The “left” part of the plot, i.e. below 0 on the log scale is therefore often used to estimate false positive interactors (FDR control). 7b shows proteins that are strongly depleted with high significance. How do the authors explain this observation? Also here, I do not support the conclusions drawn from the experimental data. The WB data shown in 7d again seems weak.

Presentation of pulldown experiments: Generally, when working with the Suppl. Tables, it is not immediately clear which Table one is looking at: Please provide the Table Number in the Excel File! In case of the Twinkle pulldown, the columns are poorly named: Figure 5a x-axis is labeled as “Twinkle / Control”, which in the Table then becomes “WT / PB”. After some digging, it becomes clear that PB stands for pBabe, which is explained once in the manuscript, however written as pB. Figure 7e is labeled with Twinkle WT / Twinkle K320E, which in the Table becomes WT / KE. In the Table, “WT / PB” and “WT_PB” are used synonymously. Similar observations can be made when comparing Figures 7a, 7b and S8b to Table S3. Some of the Figure Legends give information about the number of replicates, some don't. In the Legend of Figure 5a, the authors speak of a p-value being “< -0.05”, which was supposed to mean “< 0.05”. I am sure that more examples like these can be found throughout the manuscript. It becomes apparent, that some parts are written in a downright sloppy manner. The authors should note that this leaves the impression that the reader's time is not appropriately valued

Other data

Figures 1f, 1g: The LC3 assay, where a shift from LC3-I (soluble, cytosolic) to LC3-II (C-terminally clipped, slightly shorter, in the autophagosomal membrane) indicates a higher autophagic flux, which is then analyzed via WB w/ and w/o Chloroquine treatment (which blocks the fusion of the autophagosome with the lysosome) is apparently a well-known assay in the autophagy field. However, I guess that many readers do not know this assay. A well-written paper should try to make it easy for readers with different specialization to follow the story. However, the authors did not explain the assay anywhere in the text, no paper has been cited that explains the assay, nor have the bands have been labeled with LC-I and LC-II. In line 128-130, the authors write “However, in SOL for K320E mice, chloroquine was not inducing a change in the LC3 ratio suggesting that, in this muscle, autophagy flux was already at maximum level in steady state”. If autophagy flux w/o chloroquine was already maxed out, then I would

expect a very high ratio of LC-II to LC-I. However, this is not the case. The author should validate their point with a different assay.

Reviewer #1

In this manuscript titled “Mitochondrial membrane proteins and VPS35 orchestrate selective removal of mtDNA”, Pla-Martin et al. reveal that VPS35 and some mitochondrial proteins including Twinkle, ATAD3 and SAMM50 are involved in the remove of mutant mtDNA, and also showed that rapamycin eliminates mtDNA alterations in vivo. The findings shown in this manuscript are potentially interesting to the readers. However, the data in the manuscript are preliminary and do not strongly support some conclusions, and the mechanism of VPS35 and mitochondrial membrane proteins removing mtDNA is still obscure.

Major comments

1. How VPS35, Twinkle, ATAD3, and SAMM50 cooperate to remove mtDNA is still unclear, the pathway (mode) and the underlying mechanism are not clearly illustrated.

We show that the location and interaction of these three proteins is modified depending on the status of the mtDNA. With our work, we demonstrate that either by chemical means (EtBr) or by genetic tools (expression of K320E), the association of the nucleoid protein Twinkle with mitochondrial membrane proteins changes. In this revised version of the manuscript, we now show that, in the steady state, Twinkle associates with ATAD3, a protein which has long been known to bind to the mitochondrial nucleoid, and this association is lost upon mtDNA damage (new Fig 4e). The opposite we find for SAMM50, a mitochondrial outer membrane protein which only associates with Twinkle upon mtDNA damage (Fig. 4f). Consistent with this, we also include in this version of the manuscript a western blot showing the association of SAMM50 with K320E-Twinkle in steady state (Fig S6g).

Furthermore, in this revised version of the manuscript, we sought to investigate the area of influence and protein composition of mitochondrial nucleoids containing mutations. Thus, we are including data from proximity labelling followed by purification of wt Twinkle and K320E-Twinkle. We observed an enrichment for MOM proteins in K320E-Twinkle proteomes, in addition to endosomal vesicle trafficking proteins (Fig 4; Fig 5). Due to its implication in mitochondrial quality control, we selected VPS35 as an endosomal protein. In steady state, VPS35 can be occasionally observed in close contact to the mitochondria, however, upon mtDNA damage, VPS35 is recruited to the Twinkle enriched sub compartments (Fig 6a). In addition, we have included new data where we show that SAMM50 is necessary to deliver damaged mtDNA to VPS35 endosomes (Fig. 6c).

2. Although 8-OHdG assay is used in Fig. 2K, 2L, S4C, S4D, the assays detecting the damaged or mutant mtDNA are not enough, 8-OHdG assay and qRT-PCR assay should be performed throughout the manuscript to evaluate the integrity of mtDNA.

The expression of Twinkle missense mutations has been used as a genetic tool to induce mtDNA damage in many publications (Tynismaa et al., 2005, PMID 16301523; Baris et al., 2015, PMID 25955204; Khan et al., 2017, PMID 28768179). Recently, it has been demonstrated that the K320E mutation expressed in our mice induces multiple mtDNA rearrangements triggering mtDNA deletions and also duplications (Basu et al, 2020, PMID 33315859). As suggested by the reviewer, we have performed new analyses and quantified the relative number of lesions within 10Kb of the mtDNA using the long-run rtPCR method. This technique, described in previously published papers (i.e Zhu et al., 2017. PMID

28693618), is based in the assumption that damaged mtDNA interferes with the progression of any thermostable polymerase on the template, resulting in a decrease of mtDNA amplification when compared to non-damaged mtDNA when using the same DNA amount as a template. Thus, we have quantified the relative damage of mtDNA samples obtained from untreated cells, K320E stable cell lines, EtBr, CCCP and peroxide treated cells (see image attached). This information is presented also in figure S5C.

Quantification of mtDNA damage by qPCR

3. It is interesting that mtDNA instability induces the formation of mitochondrial protrusions, however, how mitochondrial protrusion is formed is not clear, and how mitochondrial protrusion delivers to lysosome is still unknown.

We agree with the reviewer and we will certainly address this in our future research, as it will require extensive new experimentation. However, we have new data clarifying this issue. Our new data shows that the protrusions are formed by mitochondrial membrane remodeling and they wrapped by endosomes. First, we have reconstructed in 3D the electron tomographies showing the protrusions (Fig 5a). Surprisingly, we found that some of the multimembrane structures could not be identified as mitochondrial membranes. By reconstruction, we noticed that these membranes were originally from other organelles resembling endosomes. We have also performed immunogold labelling for VPS35 and demonstrated that endosomes are recruited to the mitochondrial membrane upon mtDNA damage (Fig 6f). We also observe that in absence of SAMM50 and upon mtDNA damage, some mtDNA can be found outside the mitochondrial compartment (Fig 6c, d). We observe that mtDNA can be localize to VPS35-endosomes and that SAMM50 is required to avoid release into the cytosol. Finally, we have performed as well Correlative Light and Electron Microscopy (CLEM, Fig 6g), and visualized in late autophagy structures both Twinkle and VPS35. Our data suggests that mitochondrial fragments containing nucleoids are wrapped by endosome-like structures and mature towards late-endosome for elimination.

4. In Fig. 3, 5, S7, etc., the authors detected mtDNA removal by examining mtDNA copy number, how do the authors exclude the possibility that mtDNA biogenesis is decreased?

We detected mtDNA removal by quantifying both mtDNA copy number and dsDNA mitochondrial foci. To exclude the effect of EtBr on mtDNA biogenesis, we have performed new experiments, where we have analyzed BrdU incorporation in EtBr treated cells (Fig S4g, h). We did not observe a significant change in the number of BrdU foci. In addition, we have now analyzed the effect of EtBr in mitochondrial biogenesis by measuring the expression of *Pgc1 α* mRNA, a well-established master regulator of this process (Fig S4i). We observed a prominent increase of *Pgc1 α* mRNA suggesting that, on the contrary, mitochondrial biogenesis is not decreased, but might be even increased in order to compensate for the EtBr damage. We are also showing now a Western blot of OXPHOS index proteins, where we observed that only mtDNA encoded subunits are affected, with no changes for nuclear encoded SDHB, suggesting no changes also in mitochondrial mass (Fig S5f). We interpret that, for the time of EtBr treatment that we use in our experiment, mtDNA biogenesis is not dramatically affected and thus mtDNA depletion has a major component related to removal.

5. In Fig. 6, 7, S7, and S8, the authors used EtBr to damage mtDNA, some other mtDNA damage inducers such as H₂O₂, are required to further confirm the phenotypes.

We use EtBr as a way to damage mtDNA without affecting other mitochondrial functions, in a similar way achieved by K320E expression. In this revised version, we include a complete new set of experiments comparing the effect of other stressors such H₂O₂ and CCCP (Fig S5). In addition, as we have already explained, we have used the long-run rtPCR method to quantify mtDNA damage for all these inducers. Considering that K320E expression induces mtDNA damage and depletion of mtDNA encoded OXPHOS subunits (Complex I, III and IV, Fig S5d), without affecting the mitochondrial network, we have compared the effect of CCCP and H₂O₂. CCCP and H₂O₂ induce a strong fragmentation of the mitochondrial network but do not affect mtDNA OXPHOS encoded proteins. In addition, whereas H₂O₂ increases the relative number of lesions in the mtDNA, the effect of CCCP is mild, as expected. We conclude that CCCP and H₂O₂ affect mitochondria to a much larger extent than K320E. However, EtBr effect is very similar to the one observed by K320E expression and we believe that, using other stressors not specific for mtDNA, will result in a different outcome.

Minor comments

1. Fig. 7g and 7i, the evidence of VPS35-Twinkle contacts is weak. The selected regions for quantification should not be the middle region where a lot of mitochondria exist, leading to being hard to be distinguished by confocal microscopy.

We apologize if the quantification approach that we followed was not clear. For quantification, we use the complete cell and not a middle region. We have selected a middle region to illustrate the phenotype observed. All quantifications have been done using IMAGE J. We use a macro that we have created in which we are able to quantitate the particles of one channel present in another channel. For instance, for quantification of mtDNA foci, first we threshold the signal for all the channels. Then, we eliminate the nuclear signal (DAPI) for the dsDNA signal. In a final step, we quantify the dsDNA foci inside or outside the signal provided by TOM20 staining. The same approach has been used for VPS35 quantification in contact with mitochondria. We have clarified this now in the methods section.

2. VPS35-mtDNA-Tom20 or VPS35-mtDNA-Twinkle immunostaining can be performed to evaluate VPS35-mtDNA contacts, and check the role of VPS35 in the formation of mitochondrial protrusions.

We have performed VPS35-mtDNA-Twinkle staining. The new data is included in Fig S8f, g.

3. Line 264: Fig S5b should be Fig S6b.

We have corrected the mislabeling.

Reviewer #2

The manuscript by Pla-Martin and colleagues describes that mtDNA depletion after expressing a dominant negative version of the mitochondrial helicase Twinkle, or by treating cells with EtBr, is due to increased mtDNA turnover. Autophagy pathways were found to be important in this process but micro vesicles (MDVs) were not. The mitochondrial transmembrane proteins ATAD3 and SAMM50, participated in mitochondrial membrane remodeling required for mtDNA loss. The autophagy pathway also required the vesicular trafficking protein VPS35 which binds to Twinkle-enriched mitochondrial subcompartments upon mtDNA damage. Finally, using an in vivo muscle model, they showed that rapamycin selectively removes mtDNA deletions which accumulated during muscle regeneration, but without affecting mtDNA copy number. They concluded that mutant mtDNA elimination depends on these series of events and partners. Overall, it is a very well-done study, where the authors put a great deal of thought and effort to figure out the fate of mutant mtDNA. Said that, I have a few concerns.

1) My main concern is the interpretation of EtBr-related experiments. As far as I remember, EtBr causes mtDNA level reduction by blocking mtDNA replication. mtDNA levels are reduce because cells continue to grow during treatment, reducing mtDNA levels by dilution. Although EtBr may induce low levels of mutations, the effect or prevalence of these would be extremely low compared to the effect in replication. Therefore, to equate loss of mtDNA during EtBr treatment to active elimination by autophagy is highly debatable. If, for example the ATAD3 KD cell line was growing well, the mtDNA levels should reduce. This brings up the cell growth of the cell lines when these analyses were conducted.

We agree with the reviewer in this point and we had plenty of discussions in this regard. We have however some comments. EtBr has been classically used as a way to induce mtDNA

depletion and generate Rho0 cells. EtBr is known to reduce mtDNA replication rate and indeed, inducing mtDNA depletion. Nevertheless, taking into consideration the available protocols, the time for the cells to grow in presence of EtBr to generate Rho0 cells is always very long (from 15 days to 6-8 weeks). Here, we have used a low concentration of EtBr (50ng/ml) and the duration of our experiments never surpassed

one week. As explained for reviewer 1, we have now included a complete new set of experiments demonstrating that EtBr also damages mtDNA (Fig S5).

To answer specifically the reviewer's question, we have analyzed the growth ability of Atad3 KD cells. As can be seen in the figure, Atad3 KD cells, show a delayed growth rate compared to control in steady state, but in EtBr, both cell lines show similar delayed pattern. However, mtDNA copy number increases in Atad3 KD upon EtBr treatment (Fig 4j).

2) In Figure 1, the statement: "However, in SOL for K320E mice, chloroquine was not inducing a change in the LC3 ratio suggesting that, in this muscle, autophagy flux was already at maximum level in steady state." Is unclear. The LC3I and LC3II ratios do not appear to change in any sample, with or without chloroquine.

We agree with the reviewer and apologize for this issue. We have repeated the experiment with new reagents and increased the number of animals. As seen in the new figure 1, the autophagy flux, measured as the ration between LC3-II/LC3-I is again higher in SOL K320E. However, we did not observe a conversion from LC3-I to LC3-II in K320E mice, but a depletion of LC3-I. We agree that this dynamic is not the usual observed when canonical autophagy is increased, and hence, we have modulated our claims and suggested the activation of non-canonical pathways, which we proved later.

It is however noteworthy to mention that autophagy rate is known to be reduced in old animals and accumulation of LC3-II is not as evident as it normally is for *in vitro* experiments in cultured cells. We have tried to increase the duration of our experiment and the concentration of chloroquine, but this led to the death of the mice.

3) Regarding the use of LC3-GFP-mCherry or Fis1p-GFP-mCherry, the authors need to minimally have a sentence explaining that GFP is acid sensitive. Not every reader will be familiar with the system.

We have included a sentence clarifying how to interpret the data in this regard:

Line 178-179: "In both cases, the acidic pH of phagosomes or mitochondria, respectively, after fusion with lysosomes, induces quenching of the GFP fluorescence, facilitating visualization and quantification of autophagy/mitophagy flux"

4) In Figure 8, the regenerated cells in the rapamycin treated look very abnormal, as not only cytochrome oxidase is absent, but also succinate dehydrogenase.

We agree with the reviewer that rapamycin has some unexpected effects on tissue regeneration. We would like to emphasize that the tissue was collected only one week after muscle damage. It is known that upon cardiotoxin damage, complete muscle regeneration is achieved after 4 weeks (for detail protocol see Guardiola et al., 2017, PMID 28117768), so clearly, all physiological properties of the muscle are not yet present. It is also known that upon cardiotoxin, muscle architecture changes and a fiber shift towards glycolytic fibers IIB, with low levels of mitochondria, has been observed (for instance, recent research, Dalle et al, 2020, PMID 32621158). When we carried out COX-SDH histology, the enzymatic reaction to stain both COX/SDH was stopped to ensure a correct visualization of the entire muscle and avoid saturation. In the image presented (Fig 7), SDH staining is present, since blue cells and dark brown cells can be detected. We believe that the phenotype we observe is the cause of

a fiber shift and metabolic rewiring, produced by both rapamycin and cardiotoxin, and for this reason, we performed fiber type staining (Fig S11) and confirmed significant differences in fiber type composition.

5) This comment (and ref 22) also raised some concerns: “Paradoxically, we found that expression of the dominant negative K320E mutation of Twinkle in extraocular muscle shows a remarkable differential vulnerability of muscle fiber types, with mitochondrial dysfunction especially affecting fibers with a glycolytic metabolism (22)”. The Cre used to delete the gene (the skeletal muscle-specific Mlc1f-promoter) is much more active in type II fibers.

The reviewer is right and we are aware of the controversy about Mlc1f-promoter. However, we do not agree completely with this notion – we studied the literature concerning the choice of Cre –mice very carefully before starting these experiments: i) The MLC1f promoter is expressed even in early vertebrate somites (Theze et al., 1995; PMID 7556919), ii) It is expressed even in the slow twitch intercostal muscle (Kelly and Buckingham, 2000; PMID 10998640) – with Margaret Buckingham being a pioneer in muscle differentiation) and iii) It leads to similar weight changes in M. quadriceps, gastrocnemius and soleus when knocking out the myostatin gene

(Heineke et al., 2010; PMID: 20065166).

It has been shown that Mlc1 promoter is stronger in slow-fibers, but this does not mean that it is not active in slow oxidative fibers. The strength of a Cre-promoter reflects the chances for recombination of a specific transgene. Considering that muscle fibers are multinucleated, we understand that a “strong promoter in slow fibers”, reflects a higher number of recombination events in these fibers. In order to validate our model, we have crossed Mlc1f-Cre mice with transgenic mice carrying a mitochondrial targeted GFP. As it can be seen in the picture, GFP signal can be found in Soleus (SOL) and Tibialis anterior (TA) muscle, in the majority of the fibers. Specifically in SOL, it is true that some fibers showed a stronger signal, probably slow glycolytic fibers, but mitochondrial GFP signal could be also detected in other fibers, suggesting that Cre recombination actually occurs.

6) Finally, I believe the final model is ok, but it is presented in a very strong/conclusive manner that is not warranted by (mostly) protein-protein interaction data. I would tone it down as “the data suggests a model”.

We believe that with the new data presented after revision, the conclusions are stronger.

Reviewer #3

In this manuscript by Pla-Martin et al, the authors report that mitochondrial membrane proteins in association with VPS35 removed damaged mitochondrial DNA through a selective autophagy process that require ATG5. The authors propose that the mitochondrial DNA helicase Twinkle direct the targeting of nucleoids to the autophagic pathway in collaboration with ATAD3 and SAMM50 to remove damaged mitochondria. They propose that this process involves vesicular transport protein VPS35 but is not dependent on the formation of mitochondrial derived vesicles. There is a lot of nice work presented in this manuscript, but they seem to support the already known pathways for the removal of mtDNA and mitochondrial fragments. Moreover, there are some conclusions which are not currently supported by the data provided here. I will first make two major points and then some minor points that need to be clarified.

1. The roles of mitochondrial membrane proteins

A): Twinkle and ATAD3: The new hypothesis proposed in this work is that Twinkle may control the targeting of mtDNA to the autophagic pathway. However, results showed so far do not support this conclusion. ATAD3B has already been reported to regulate the removal of damaged mitochondrial DNA (Shu et al, 2021; PMID: 33665835). Is Twinkle required for the removal of mtDNA? Results here do not support a role for Twinkle but only support the already known role of ATAD3 in clearance of mtDNA. To ascertain if Twinkle have any role(s) in mtDNA clearance, you will need to clarify if Twinkle interaction with ATAD3 is necessary and sufficient for mtDNA clearance or show that the process described here is not the same as the one already reported for ATAD3?

It was not our intention to study new functions for Twinkle, but to use it as a tool to mark nucleoids or to modify them. We apologize if the message was not clear. To assay what the reviewer is suggesting, we would need to generate models where Twinkle is unfunctional or removed, and assay there mtDNA turnover. Unfortunately, generation of a Twinkle KO model is not possible. It has been attempted in mouse, leading to early embryonic lethality, but it was possible in conditional models, where it led to drastic mtDNA depletion (Milenkovic et al., 2013; PMID: 23393161).

However, we present here that the interaction pattern of Twinkle, used as a way to label mitochondrial nucleoids, with two mitochondrial membrane proteins, ATAD3 and SAMM50, changes depending on the presence or absence of mtDNA damage. Nevertheless, co-immunoprecipitation experiments cannot prove if an interaction is direct or other nucleoid proteins are mediating it but, the fact that these proteins can be pulled down together, suggests that at least they are in a protein complex. It is worth to mention that recently, Sam50 has been shown to form MICOS-ATAD3-mtDNA axis, controlling mtDNA stability and cardiolipin remodeling, related to mtDNA release (Chen et al., 2022. PMID 35313046). These results are completely in line with our findings. However, here we prove that ATAD3 and SAMM50 bind Twinkle depending on the state of the mtDNA.

In addition, to answer reviewers' question regarding ATAD3, it is important to mention that we have worked with mouse derived cells (C2C12 and MEFs), where only one *Atad3* gene is present. In humans, *ATAD3A* and *ATAD3B* were generated possibly by a gene duplication (Gunning et al., 2020, PMID 32004445). It is possible that both genes have evolved to acquire

more specific roles in higher organisms. And in fact, very recently, the role of ATAD3A in membrane architecture was clarified (Arguello et al., 2021, PMID 34936866).

B): Twinkle and SAMM50. Does Twinkle bind directly to SAMM50 and is this interaction relevant for clearance of mtDNA?

As we have already discussed, co-immunoprecipitation (co-IP) experiments cannot prove a direct interaction between two proteins, but association between them. Based on our co-IP experiments, we cannot exclude those other proteins are mediating Twinkle-SAMM50 association. In this revised version of the manuscript, we present also a co-IP experiment between SAMM50 and K320E-Twinkle, and show that the association between them is also possible in steady state (Fig S6g).

However, we would like to emphasize that in this version, we now are including Proximity Biotinylation experiments with APEX2. Upon H₂O₂ damage, APEX2 catalyzes biotin-phenol crosslinks to proteins in close proximity (Hung et al., 2016, PMID 26866790). This reaction is specific for the subcellular compartment and allows the identification of weak and transient protein associations. We and others (Han et al., 2017, PMID 28238724) have seen that, upon wt-Twinkle-APEX2 proximity biotinylation, the majority of the enriched mitochondrial proteins are located in the mitochondrial matrix. However, when we use K320E-Twinkle-APEX2, we observe an enrichment of MOM proteins, including SAMM50, VDAC1, VDAC2, VDAC3 and several more. This enrichment would not be possible if K320E-Twinkle was still located in the same location as the wt-Twinkle, suggesting some membrane reorganization which allows mutated Twinkle to associate with MOM proteins.

SAMM50 is a beta barrel protein that is localized in the outer mitochondrial membrane. It is known to interact with the TOM complex and assemble both the MICOS and SAM complexes. SAMM50 has also been reported to control basal removal of mitochondrial fragments without affecting the mitochondrial network (Abudu et al, 2021; PMID: 34037656). If Twinkle have roles in mtDNA removal through interaction with SAMM50, you will have to show how and the relevance of this interaction during this process. Is the SAM and MICOS complexes also involved?

With proximity biotinylation experiment, we detected several proteins for the MICOS complex such *Immt* (MIC60). We have attempted a direct coIP between Twinkle and MIC60 but this was not successful (see image attached). We believe that MICOS proteins are involved in the process, as they could be detected in the area of influence of K320E. Unfortunately, we do not have evidence of a strong interaction between Twinkle and other MICOS complex proteins besides SAMM50.

How do you separate the role of SAMM50 in basal piecemeal mitophagy from the clearance of mtDNA or are these the same process?

With the new data we present here (Fig 6), we show that SAMM50 is necessary to locate dsDNA to VPS35 endosomes. In the discussion, we comment on the role of SAMM50 in

piecemeal mitophagy. Our data supports that at least mtDNA damage does not trigger canonical mitophagy but a specialized form, linked to endo-lysosomes. VPS35, SAMM50 and ATAD3 have been already identified during PINK-PARKIN mitophagy. We have observed colocalization of K320E-Twinkle with LC3 and LAMP1, which suggests that this process might be representing a specialized form of mitophagy which is triggered by mtDNA damage. We believe that the process we are observing is similar to piecemeal mitophagy, as only specific parts of the mitochondrial membrane containing Twinkle and mtDNA are removed and we comment on this in the discussion.

C). How does SAMM50 interact with VPS35 and is this relevant for this process. What about VDACs and TOM40 identified in VPS35 IP?

In the previous version of this paper, we showed enrichment of mitochondrial proteins upon VPS35 IP and EtBr treatment. We quantified this and observed that the enrichment of mitochondrial proteins was significant. However, we noticed that EtBr triggered also unspecific binding of mitochondrial membrane proteins to Protein A/G-magnetic beads, even in negative control samples, which was producing a higher background. Therefore, and as suggested by reviewer 4, we repeated again all the experiments using Protein A-Agarose beads but, unfortunately, we could not replicate the same result in regards of MS, and we have eliminated these data. We could only replicate SAMM50-VPS35 pull-down (Fig S10a). As it is known for vesicle trafficking, vesicle membrane proteins including VPS35 interact with other membrane protein in a process known as kiss-and-run. The interactions among these proteins are usually transient and weak and improved purification techniques are required to detect them. Our data supports that VPS35 endosomes come in close contact to mitochondria only upon mtDNA damage and show a prominent role of SAMM50. Other proteins such VDACs and TOM40 are now found in proximity biotinylation of K320E nucleoids, but if these proteins interact with VPS35 directly and to which extend, it is unknown, and we will address it in the future.

2. Roles of mitochondrial derived vesicles

A): The data provided here does not demonstrate whether mtDNA degradation occur through another pathway independent of MDV formation. VPS35-retromer complex is well known to regulate MDV formation and seem to be important in mtDNA degradation shown here. Is the mechanism of MDV formation by the VPS35 retromer complex different from the formation of these mtDNA containing fragments?

Can we separate the role of VPS35 in MDV formation from the one you describe here? Are the components or cargoes of MDVs different from the components of the mtDNA fragments? Mutational analysis (on VPS35) is needed for the former while CLEM studies are required for the latter to properly differentiate these. The absence of MAPL or TOM20 or DRP1 is not sufficient to conclude that these fragments are not MDVs.

VPS35 has been linked to non-canonical mitophagy through MDVs, where it initiates the force to generate a vesicle. The E3 Ubiquitin ligase MAPL has been shown to direct VPS35-mitochondrial cargo to peroxisomes, whereas the endosomal adaptor Tollip specifically divert these MDVs to lysosomes. We checked the E3 Ubiquitin Ligase MAPL and showed that indeed, MDVs generated by MAPL overexpression do not colocalize with Twinkle or VPS35 (Fig S9). In

addition, experiments with the lysosomal inhibitor chloroquine (CQ), suggest that VPS35 is directed to degradation (Fig S8h,i and S9). It is noteworthy to mention that recently, the proteome of MDVs has been described and a possible location of mtDNA in MDVs has been discarded (König et al., 2021, PMID 34873283).

As suggested by the reviewer, we have performed CLEM (Fig 6g). We observed that upon EtBr treatment, Twinkle localizes together with VPS35 in late autophagy structures. In addition, we have performed immunogold labelling of VPS35. In this case, we have never observed formation of MDV or translocation of VPS35 to the mitochondria, but endosomal recruitment in close proximity to the mitochondria upon mtDNA damage (Fig 6f). We believe that mtDNA removal involved endosomal recruitment in close proximity to the mitochondrial sub-compartments containing nucleoids, and VPS35 is involved in the maturation of these endosomes, but this process is independent of the MDVs machinery (Fig S9).

B): Another question is what does these fragments or particles contain and how are they shuttle to the lysosome or how are they recruited to autophagosomes? Electron tomography pictures (Figure 4) show that these fragments contain multi-membrane structures and not just nucleoids. Do these fragments contain inner mitochondrial membranes or proteins? Does it contain OMM or proteins? Are these fragments different from those degraded by SAMM50 mediated basal mitophagy (Abudu et al, 2021; PMID: 34037656) which degrades mitochondrial fragments independent of canonical mitophagy.

For APEX2 electron microscopy, the cells were treated for 1 min with H₂O₂, which triggers DAB deposition. During EM sample preparation, DAB stimulate Osmium precipitation and generates contrast. The APEX2 technique has been used in the past to visualize mitochondrial nucleoids (Han et al., 2017, PMID 28238724). Therefore, we know that the black precipitate observed in our images (Fig S8a) is closely associated with Twinkle and therefore a nucleoid. The signal detected in K320E-APEX2 cells is restricted to areas with reorganized membrane structures. We have observed similar features in wt Twinkle-APEX2 cells after EtBr treatment (see attached image). We believe that this is a common phenomenon upon mtDNA damage.

Our new data with proximity biotinylation enrichment (Fig. 4) suggest that these areas are enriched with MOM proteins, but we do not know yet if these proteins also follow the degradation route or are just performing a structural role. We have now reconstituted in 3D some electron microscope tomograms (Fig 5a). We observe that some of the membranes are originally derived from other organelles in close proximity to mitochondria, which we have identified as endosomes. As we discussed above, we believe that this process is similar to the piecemeal mitophagy, and at least nucleoids containing Twinkle are included here. To detect if other mitochondrial proteins follow this pathway upon mtDNA damage, further extensive experimentation will be required.

Another important question is how to explain the importance of ATG5. Some specific MDV delivery to the lysosome is shown to be independent of LC3 and ATG5 (Soubannier et al., 2012; PMID: 22226745). The question here is how does these fragment fuse with ATG5-dependent autophagosome or how are they shuttle t the lysosome. Is the ATG8 proteins involved?

As we discussed in the paper, we believe that this process is independent of MDV. As explained before, mtDNA has not been found in association with MDVs (König et al., 2021, PMID 34873283). We selected ATG5 because it has been involved in unconventional forms of mitophagy (Codogno et al., 2011, PMID 22166994). We understand reviewer's concern, as ATG8 has been also involved in unconventional autophagy pathways (Durgan et al., 2021, PMID 33909989). In our case, we disregard if other ATGs are involved, however, we will be happy to address this in the future. Here we intend to give an overview of a completely new process linked to specific mtDNA degradation, and the validation of other ATGs proteins will complicate and conceal other information, which we believe is more within the scope of our research. We can hypothesize that ATG5 might have a role in the endosome-mitochondria membrane binding, as it was described previously (Romanov et al., 2012, PMID 23064152) but further experiments will be required.

However, as suggested by this reviewer below, we have performed new experiments using autophagy blockers to verify the involvement of autophagy in mtDNA depletion (Fig 2h, Fig 3g). We thank the reviewer for bringing up this question, because we actually observed that neither 3MA nor SBI0206965 recover mtDNA depletion induced by K320E-Twinkle expression. Only Chloroquine is effective in doing so, suggesting that lysosomal function but not general autophagy is involved in the process.

Minor Points

1. Expression of the K320E mutants seem to lead to altered mitochondria morphology (Fig S3) Can you quantify mitochondrial morphology.

We have quantified mitochondrial morphology for K320E cells. We observe a minor effect produced by K320E expression (Fig S5a, b)

2. It is now well known that OMM and IMM/Matrix proteins are degraded differently during mitochondrial quality control. Can you blot for IMM/Matrix protein as part of Fig. 1

We have included blots to complement Fig 1. As observed in Fig S2a-c, only mitochondrial outer membrane proteins accumulate after CQ treatment in SOL from K320E mice, which suggest a different turnover rate for these proteins.

3. Does expression of K320E drives basal mitophagy. Figure 2C is difficult to interpret as lack of mitophagy. Use Matrix targeted Keima or- tandem tag to show basal mitophagy levels.

We have indeed used a tandem plasmid named Fis1p-Tandem (GFP-mCherry). This tool has been used multiple times to monitor mitophagy flux (for instance, Allen et al., 2013, PMID 24176932; Pla-Martín et al., 2020, PMID 32149416). We have used Manders'coefficient, an index that quantifies the colocalization of one dye to another one, which is independent of

signal intensity (Manders et al., 1993, PMID 33930978). Thus, mitophagy will be detected as reduced Mander's coefficient, due to lack of colocalization of red spots (derived from mitophagosomes) with normal mitochondria (red and green signal). We have now included a representative image of cells treated with antimycin/oligomycin, which is known to trigger canonical mitophagy.

4. Explain how you quantified nucleoids and describe in detail (preferably in materials and method) how you quantified mtDNA copy number. It is difficult to reconcile reduction in mtDNA copy number with no changes in nucleoids.

We apologize if the quantification method was not clear. As explained for reviewer 1, we use the complete cell and all quantifications have been done using IMAGE J. Using a macro that we have created, we are able to quantify the signal of one channel present in another channel. For instance, for quantification of mtDNA foci, first we threshold the signal for all the channels. Then, we eliminate the nuclear signal (DAPI) for the dsDNA signal. In the final step, we quantify the dsDNA foci inside or outside the signal provided by TOM20 staining. A similar approach has been used for VPS35 quantification in contact with mitochondria. All these methods are detailed in the M&M section – "Image analysis".

Regarding the question about nucleoids and mtDNA copy number, it is known that a mitochondrial nucleoid contains on average, 1.5 molecules of mtDNA and a size of 80x80x100nm (Kukat et al., 2015, PMID 26305956). Further, in this work, we have used traditional confocal microscopy, in which the resolution limit is restricted to 200nm. Hence, it is possible that mtDNA foci in confocal pictures contain more than 1 nucleoid, as it was demonstrated previously (Silva Ramos et al., 2019, PMID 31170154). Hence, a reduction of mtDNA copy number does not always correlate with a reduction in foci number in traditional confocal microscopy.

Can you stain with mtDNA antibody and then do quantification in all these conditions?

We question whether a specific antibody against mtDNA exist. We use an antibody against double strand DNA (dsDNA) which specifically stains nuclear and mtDNA. We could have used an antibody against a nucleoid protein but this would have been out of the scope of our research. Our antibody is used in the mitochondrial community to stain mtDNA, and all the quantifications have been done using it. We hope that the reviewer understands that demanding the repetition of all the experiments with another antibody or a mitochondrial nucleoid marker will be very time consuming and will not necessarily lead to new data. We intended to damage mtDNA, either with EtBr or K320E-Twinkle expression, and we follow mtDNA dynamics, as we sought to discover new quality control pathways for the mtDNA.

5. You showed that K320E expression reduced the mtDNA copy number in both WT and ATG5 KO cells, but the nucleoid number do not change in ATG5 KO pointing to an ATG5-dependent removal of nucleoids. However, the copy number can be rescued in ATG5 WT with chloroquine treatment cells but not in KO cells. This statement is contradictory. This implies that mtDNA copy number is reduced in lysosomal-dependent manner but not ATG5 dependent manner, while nucleoids are removed in ATG5 dependent manner. Can you explain this.

What we observe here is a dual role of K320E on mtDNA depletion. In replicative cells, expression of K320E leads to mtDNA depletion (Weiland et al, 2018, PMID 28867657; Holzer et al., 2019, PMID 31085560) and we also observe it here, for C2C12 cells (Fig 2e) and for Atg5 KO MEFs (Fig 3c). We demonstrate here that mtDNA depletion caused by K320E in wt cells (C2C12 and ATG5 wt) is lysosomal dependent, as it can be recovered by chloroquine (Fig 2h and Fig 3f). However, in Atg5 KO cells, we observe that K320E decreases number of mtDNA replication foci stained by BrdU (Fig 3h, i), suggesting that in this case, mtDNA depletion is mediated by mtDNA replication delay. To overcome this issue, we tried to find other ways to damage mtDNA in a similar manner to K320E expression and used EtBr instead (Fig S4 and S5). By using EtBr, we observe that mtDNA depletion following mtDNA damage was inhibited in ATG5 KO cells, suggesting that it is indeed ATG5 dependent (Fig 3h-j).

6. You then suggested that Loss of ATG5 seems to affect mtDNA replication during K320E expression. Shouldn't mtDNA replication affect mtDNA copy number? If ATG5 KO (or autophagy) affects mtDNA replication, it should also affect mtDNA copy number. Can you explain this? Is this specific to ATG5 or is it autophagy per se?

However, Ethidium bromide treatment leads to reduction of both nucleoids and mtDNA copy number in WT cells and not ATG5 KO cells. Why is this different from K320E expression? Are these two different processes?

Unfortunately, we do not understand reviewers' concern because this is exactly what we show in Fig 3. We have explained above this issue.

Line 213-215: "Our data suggest that in *Atg5* deficient cells mtDNA depletion is caused by reduced mtDNA replication, presumably to avoid accumulation of excessive mtDNA damage. In contrast, in wt cells, K320E triggers mtDNA depletion linked to an increased turnover rate."

7. Alternate to ATG5, can you use inhibitors to ULK complex or PI3K complex or another ATG KO (FIP200, ATG7 or ATG9) cells to corroborate the role of autophagy.

We have now used 3MA and SBI0206965. 3MA is known to inhibit autophagy by blocking autophagosome formation, while SBI0206965 is a highly selectivity ULK inhibitor. None of these two inhibitors have an effect on recovering mtDNA after depletion (Fig 2h and Fig 3g), suggesting that in fact canonical autophagy is not involved in this process. However, as Chloroquine was able to recover mtDNA copy number levels and rapamycin was able to ameliorate mitochondrial dysfunction caused by mtDNA damage, we believe that an endo-lysosomal pathway is involved here.

8. i. In Fig3a, why is the pBabe picture is also positive for V5 staining?

The picture presented in Fig 3a is not positive for V5 staining and this represents a normal background observed in many immunofluorescence pictures, produced by unspecific binding of V5 antibody to similar protein epitopes in the cell. That signal is not comparable to the intensity and resolution observed in cells transduced with

Twinkle-V5. To convince the reviewer, we present here only the V5 and DAPI channels for that panel

- ii. **Change Fig. S4c,e to S4e,f; line 193**
- iii. **Check Graph labeling for Fig. 3L**
- iv. **Change Fig3d to 4d; line 220**
- v. **Change Fig S5b to S6b; line 264**
- vi. **Change Fig. S8a,b to S9a,b; line 325**
- vii. **Change Fig. S8c, to S9c; line 326**

In this revised version, we have modified the order of the figures. We have verified that each figure is properly labelled and indicated throughout the text.

Reviewer #4

Pla-Martin et al report on a new, autophagy-dependent recycling pathway for the removal of mutated/altered mtDNA, involving the mitochondrial DNA helicase Twinkle, the inner membrane protein ATAD3, the outer membrane protein SAMM50 and the vesicular trafficking protein VPS35. In their experiments, they make use of the disease-associated Twinkle variant K320E, which causes mtDNA alterations. They study the resulting cellular recycling responses in vivo (mouse muscle tissue) and in vitro (C2C12 cells and MEFs). Methodologically, they employ a wide variety of approaches, including (q)PCR, WB, Fluorescence microscopy, electron microscopy and proteomics.

I acknowledge the amount of work invested in this study and I believe that if rigorously done, the findings claimed here would be of value for the scientific community. However, I do not see this manuscript as a suitable candidate for Nature Communications due to the below detailed reasons / comments. In brief, the presented data does not (sufficiently) support the draw conclusions. To my opinion, the manuscript does not convey information in a clear and coherent manner (inconsistencies across figures, suppl tables and main text). Due to my own area of research, I mainly focus on the presented proteomics data.

Proteomics data

pSILAC (Figures 1h, S2): State of the art pSILAC experiments, whether done in cell culture (see PMID: 29414762) or in mouse model (see PMID: 30315172), track label incorporation over several timepoints. Based on curve fits, these timepoints serve as an additional quality control layer and allow to (quite accurately) calculate protein turnover rates. These turnover rates can then be compared across samples and between proteins. Including a label-switch (chasing e.g. 2 Reps from L to H and 2 Reps from H to L) into the experimental layout further increases the reliability of the data. Here, the authors only used 1 timepoint, which delivered H/L ratios. In order to compare WT to KE across muscle samples, the authors calculated ratios of these H/L ratios, which amplifies inaccuracies in quantification. Also, a label-switch has not been done, although I do acknowledge that this is much easier done in cell culture than in mouse experiments. Finally, simply doing a student's t-test is no longer state of the art.

Instead, some kind of FDR-correction should be included (giving rise to adjusted p-values, termed q-value by MaxQuant / Perseus; see also below). By applying these improvements, the results would be much more reliable. The conclusions that can be drawn from the pSILAC data presented here are limited.

The aim of the pulsed SILAC experiment was the quantification of L-¹³C₆-lysine (heavy lysine) incorporation into newly synthesized proteins after two weeks of feeding a heavy lysine-containing diet. We assumed a 14-day feeding experiment as sufficient to cover heavy lysine incorporation into a broad range of newly synthesized proteins. A recent study showed a median protein half-life of eight days in mouse skeletal muscles (Rolfs et al., 2021. PMID: 34836951) supporting the chosen timespan. Our data indicate that independent of muscle or phenotype no changes in heavy lysine incorporations can be observed. This is not only highlighted by the lack of significantly changed H/L ratios of proteins identified between Twinkle-WT and Twinkle-K320E animals but further by the similar average incorporation rates as indicated in figure S2e. Also, while different muscles have different incorporation rates as indicated by a principal component analysis (Fig. S2d) this is not the case between phenotypes.

We acknowledge that the readout of this experiment is limited since we only generated one timepoint. In the manuscript we now specify that only incorporations were determined to avoid false conclusions by the reader. We further reanalyzed all proteomics data and used only FDR-corrected q-values in combination with log₂-transformed fold changes to identify significantly changed heavy lysine incorporation rates or enriched proteins between Twinkle wild-type and Twinkle-K320E conditions.

Twinkle pulldown (Figures 5a, 7e): For AP-MS experiment, the authors relied on label-free quantification using MaxQuant. As already mentioned for pSILAC, a standard t-test p-value is used instead of the adjusted q-value. However, the authors provide q-values in Table S2. When requiring a q-value of <0.05 for significance, which is more appropriate, ATAD3 is no longer significantly enriched and the only proteins that remain significant are Peo1, Rps29 and Dlat. In Figure 7e, no protein remains significant based on q-values. In lines 235-237, the authors write: "As expected, the majority of proteins interacting with Twinkle were related to mtDNA replication, transcription and translation, but only a few interactions were found to be significant". This claim is unproven, the authors should show this via a GO-Enrichment analysis. Nevertheless, an enrichment of non-significant proteins is not reliable. About the ratio of enrichment (and ignoring statistics) for Atad3 and Samm50 (5a and 7e, respectively): With ~4-fold enrichment, Atad3 could be a genuine (weak) interactor, which is also shown in Figure 5b.

We apologize for the inconvenient statistical analysis of proteomics data. As suggested by the reviewer we reanalyzed all proteomics data and used exclusively FDR-corrected q-values in combination with log₂-transformed fold changes between conditions to identify significant hits. The commented sentence was removed since we were not able to prove this statement by GO enrichment.

We further included proximity biotinylation experiments using APEX2-tagged Twinkle-WT and Twinkle-K320E constructs. By this we statistically proved increased interactions of K320E with mitochondrial membrane proteins which was proven by direct comparison of K320E with WT as well as by GO annotation enrichment analyses. The results of the proximity biotinylation

experiments are visualized in figure 4 and S6. The heat maps of Euclidean hierarchical clustering analyses highlight the enrichment of mitochondrial inner and outer membrane proteins when expressing Twinkle-K320E.

We further include a reanalyzed version of the already shown affinity enrichment experiments of Twinkle-WT. Instead of downshift missing value imputation deterministic minimum value imputation was used. Also, for other AP-MS experiments the same imputation method was used. In combination with an improved statistics (q-values and log2 fold changes) the interaction between Twinkle-WT and Atad3 was clearly shown.

However, in 5b, the authors should add a mitochondrial control protein that is not enriched in order to prove their point.

As explained for reviewer 3, MIC60 does not IP with Twinkle, which would serve also as a negative control for this reviewer's question.

For Samm50 (Figure 7e), with an almost 8-fold difference, the protein is of much higher abundance in the KE mutant. However, looking at Table S2, this is solely due to a very low abundance in the WT. When comparing KE to the Control sample, less than a 2-fold enrichment can be observed. Samm50 is more than 4-fold less abundant in the WT than in the empty vector control. How do the authors explain this observation? This also contradicts 5f. In 5f, the pulldown data generally looks quite weak. Taken together, to me, an Atad3 interaction seems possible/likely, albeit weak, for Samm50, the data is not convincing.

Vps35 pulldown (Figures 7a, 7b, S8b): Here, the authors appropriately use the q-value for statistics (why here and not before?). What is concerning, however, is that significant proteins are about as frequently depleted as they are enriched (see for instance 7b). This is somewhat atypical for an AP-MS experiment, because only interactors should significantly differ from to the control. The "left" part of the plot, i.e. below 0 on the log scale is therefore often used to estimate false positive interactors (FDR control). 7b shows proteins that are strongly depleted with high significance. How do the authors explain this observation? Also here, I do not support the conclusions drawn from the experimental data. The WB data shown in 7d again seems weak.

Instead of AP-MS experiments we performed proximity biotinylation experiments with Twinkle-WT-APEX2-V5 or Twinkle-K320E-APEX2-V5 protein constructs to evaluate spatial differences of wild-type and mutant interactors in the mitochondrion. A mitochondrial matrix-targeted construct (mitoAPEX) served as a control. Among the sets of potentially biotinylated proteins Twinkle-K320E showed an increased prevalence to mitochondrial membrane proteins including SAMM50. This was both shown by Student's t-testing using q-values, by 1D annotation enrichments, and by Euclidean hierarchical clustering of significantly different, mitochondrial annotated proteins between Twinkle-WT and K320E (Fig 4c). We hope that with this new data, we can convince reviewer 4 for the true nature of Twinkle-K320E interaction with SAMM50 and other mitochondrial membrane proteins.

We agree with the reviewer's comment that in AP-MS experiments only enriched proteins should be detected. We affinity-purified endogenous VPS35 using a monoclonal antibody. Endogenous IPs often suffer from high levels of background binding proteins that interact with the antibody backbone rather than with the antigen. This could explain the comparably high amounts of "depleted" proteins, which were enriched in the IgG beads control. However,

to avoid controversy, we have eliminated this data set but proved our point using other techniques.

Presentation of pulldown experiments: Generally, when working with the Suppl. Tables, it is not immediately clear which Table one is looking at: Please provide the Table Number in the Excel File! In case of the Twinkle pulldown, the columns are poorly named: Figure 5a x-axis is labeled as “Twinkle / Control”, which in the Table then becomes “WT / PB”. After some digging, it becomes clear that PB stands for pBabe, which is explained once in the manuscript, however written as pB. Figure 7e is labeled with Twinkle WT / Twinkle K320E, which in the Table becomes WT /KE. In the Table, “WT / PB” and “WT_PB” are used synonymously. Similar observations can be made when comparing Figures 7a, 7b and S8b to Table S3. Some of the Figure Legends give information about the number of replicates, some don't. In the Legend of Figure 5a, the authors speak of a p-value being “< -0.05”, which was supposed to mean “< 0.05”. I am sure that more examples like these can be found throughout the manuscript. It becomes apparent, that some parts are written in a downright sloppy manner. The authors should note that this leaves the impression that the reader's time is not appropriately valued

We greatly apologize for the non-sufficient and misleading labeling of figures and supplemental tables. It was not our intention to give readers a feeling to waste time reading our manuscript. After revising figures and tables we now hope that it is much easier to understand figures by precise legends and find the corresponding data in the respective supplemental tables.

Other data

Figures 1f, 1g: The LC3 assay, where a shift from LC3-I (soluble, cytosolic) to LC3-II (C-terminally clipped, slightly shorter, in the autophagosomal membrane) indicates a higher autophagic flux, which is then analyzed via WB w/ and w/o Chloroquine treatment (which blocks the fusion of the autophagosome with the lysosome) is apparently a well-known assay in the autophagy field. However, I guess that many readers do not know this assay. A well-written paper should try to make it easy for readers with different specialization to follow the story. However, the authors did not explain the assay anywhere in the text, no paper has been cited that explains the assay, nor have the bands have been labeled with LC-I and LC-II. In line 128-130, the authors write “However, in SOL for K320E mice, chloroquine was not inducing a change in the LC3 ratio suggesting that, in this muscle, autophagy flux was already at maximum level in steady state”. If autophagy flux w/o chloroquine was already maxed out, then I would expect a very high ratio of LC-II to LC-I. However, this is not the case. The author should validate their point with a different assay.

As explained for reviewer 2, we apologize for this issue. We have repeated the experiment with new reagents and increased the number of animals. We have also tried to explain the experimental approach for non-autophagy readers. However, we find our approaches sufficient to validate the increased autophagy flux. These experimental approaches are the gold standard for analyzing autophagy flux (Klionsky et al., 2021. PMID 33634751). Other methods would involve generating new mouse models, for instance, an LC3-GFP mouse crossed with our K320E^{skm}, and analyze data after 2 years.

Nevertheless, we noticed that the LC3-II/LC3-I ratio increases because of a reduction in the cytosolic form LC3-I more than for an increase in autophagosome LC3-II, which might be due to the non-canonical form of autophagy we are describing in this manuscript. However, it is too early to speculate that and we wouldn't like to make a strong statement there, when we actually don't have enough data for that. Our conclusions are obtained after several experiments: i) we find reduced protein levels of LC3 and p62 in the steady state from soleus protein extracts; ii) we find reduced puncta/fiber for LC3 and p62 only in soleus and iii) LC3-II/LC3-I ratio is increased only in soleus of mutant mice. In addition, we have found consistent results *in vitro*, when we have expressed the tandem plasmid LC3-GFP-mCherry in cells transduced with K320E-Twinkle.

REVIEWER COMMENTS

Reviewer #1 (Remarks to the Author):

The revised manuscript quality has been improved, and most of my concerns were addressed and resolved. However, it is still unclear how mitochondrial nucleoid containing mtDNA gets through mitochondrial inner membrane upon mtDNA damage; In Figure 6h, the authors provide a proposed model that shows ruptured (broken) mitochondrial inner membrane upon mtDNA damage. Therefore, how does mtDNA damage induce the rupture of mitochondrial inner membrane? The evidence and discussion is needed.

Reviewer #2 (Remarks to the Author):

In the revised manuscript, Pla-Martin and colleagues addressed the reviewers' many concerns. One issue that I still have concerns is the EtBr interpretation. Analysis of mtDNA damage by long template amplification is tricky. The raw data is not shown and the lower amplification may be related to the one week treatment-related mtDNA depletion. The additional treatments with H₂O₂ and CCCP were good additions, but as discussed, they have pleiotropic effects.

Reviewer #3 (Remarks to the Author):

In this revised manuscript by Pla-Martin et al., the authors have added new data and addressed some of my concerns.

However, how ATAD3 and SAMM50 acts in this pathway is not described. What does these proteins really do? Is their role(s) in this pathway the same as already reported for ATAD3 and SAMM50? Moreover, SAMM50 have important role in mitochondrial membrane protein insertion and cristae architecture. The effect on mtDNA clearance may be an indirect effect of SAMM50 in membrane protein integration and cristae stability. SAMM50 and ATAD3 involvement in this pathway as stated in this manuscript was determined through their interaction with Twinkle. I asked in my first review if their interaction with Twinkle is relevant for their role in mtDNA clearance. You have not clarified this. This is quite simple, map out their binding to Twinkle using the co-IP experiments. If this is not possible because they do not actually bind directly, use mutants already shown to act in the already described pathways for ATAD3

(mitophagy of mtDNA) and SAMM50 (piecemeal mitophagy) for reconstitution in ATAD3 and SAMM50 KD cells to rescue mtDNA clearance. This will give a clearer understanding if this is a different pathway or those already described.

In addition, the authors make too generalized statements regarding MDVs, and this needed to be modified. For example, the statement in line 524 to 525 “In addition, recently, it has been demonstrated that mtDNA cannot be localized inside of MDVs” is misleading and should be modified. I advised the authors to read the paper by König et al., 2021, PMID 34873283 properly. König et al., 2021 described the proteome of steady state TOM+ MDVs, and they stated that these MDVs may be different from stress/signal induced MDVs and other classes of MDVs. They then stated that steady state TOM+ MDVs do not contain mtDNA because the presence of mtDNA could not be validated, so is the presence of GTPases MFN1 and OPA1. We do not yet know if other classes of MDVs such as stress/signal induced MDVs contain mtDNA. The current understanding is that there are several classes of MDVs, and the proteome of stress/signal induce MDVs and other classes of MDVs including TOMM20-, PDH+ and MAPL+ MDVs have not yet been described. What you report here seem to be stress induced mtDNA removal in VPS35 positive vesicles and this may well be different from steady state TOM+ MDVs described by König et al., 2021. But to say mtDNA cannot be localized to MDV is incorrect. There is yet no proof for that. Since you only tested MAPL-induced MDVs, you should state that this pathway is independent of MAPL-dependent MDVs.

Minor comments

There is no description in the legend for figure S9E and S10D (change S10E to S10D). Check the other legends properly.

Reviewer #4 (Remarks to the Author):

In this revised version, I now believe that the presented MS experiments, together with supporting WB data, are largely sound and support the drawn conclusions.

Results section 1 (Figure S2):

Protein incorporation rates are comparable between K320E and WT, no enhanced turnover of mitochondrial proteins in KE mutants

Results section 3 (Figures 4, S6):

More proteins with vesicular (and OM) annotation are enriched in KE-APEX compared to Twinkle-APEX.

SAMM50 interacts with Twinkle-KE, but not with Twinkle-WT (unless adding EtBr), i.e. SAMM50 interacts with Twinkle under mtDNA damaging conditions.

Opposite is true for ATAD3, which weakly interacts with Twinkle-WT, but loses interaction under EtBr conditions.

Q: Why do these two proteins behave differently? Do you have a hypothesis for this observation?

Please check whether ATAD3 is really significant in Figure S6f middle plot. I have a hard time believing that given that it is in a cloud of non-significant proteins.

Please correct typo Figure S6f: q-value < -0.05

Taken together and viewed from the MS angle, I believe the revised version is more coherent. However, looking at the manuscript overall, I still believe that the presented data does not fully support some conclusions. For example, I am a bit hesitant to agree that the presented data clearly shows the proposed role of ATAD3.

REVIEWER COMMENTS

Reviewer #1 (Remarks to the Author):

The revised manuscript quality has been improved, and most of my concerns were addressed and resolved. However, it is still unclear how mitochondrial nucleoid containing mtDNA gets through mitochondrial inner membrane upon mtDNA damage; In Figure 6h, the authors provide a proposed model that shows ruptured (broken) mitochondrial inner membrane upon mtDNA damage. Therefore, how does mtDNA damage induce the rupture of mitochondrial inner membrane? The evidence and discussion are needed.

We agree with the reviewer that this is an important point and we are definitely eager to define it. However, a deep understanding of how mitochondrial nucleoids are translocated from the mitochondrial matrix to the endosomal compartment would require extensive research and years of further work. Still, we can now provide new evidence which will direct our future research. As we already discussed, we hypothesized that mtDNA elimination should be followed by mitochondrial membrane rearrangements. Recently, Chen et al., 2022 (PMID 35313046) showed that SAMM50 and ATAD3 coexist in a complex regulating cristae morphology. In this revised version of the manuscript, we show that SAMM50 and ATAD3 can be co-immunoprecipitated. Moreover, this interaction is inhibited upon EtBr treatment damaging mtDNA (Fig 4h, i). We found that cells in which ATAD3-levels have been decreased also contain low levels of SAMM50 and vice versa, confirming the data observed by Chen et al. (Fig S8c). To further characterize the mitochondrial changes induced by mtDNA damage, we have performed STED microscopy and now visualize mitochondrial cristae. Our data shows that EtBr treatment dramatically changes the inner membrane architecture. A similar morphology can be observed in cells expressing K320E-Twinkle (Fig 4j, k and S8f). In addition, we have measured changes in mitochondrial membrane potential by TMRE. Super-resolution microscopy using an Airy Scan microscope showed that both EtBr or expression of K320E-Twinkle induce local rearrangements of the mitochondrial inner membrane potential. However, gross cellular mitochondrial membrane potential measured by Flow Cytometry does not show large changes upon EtBr treatment or in cells expressing K320E-Twinkle (Fig. S7)

We agree with the reviewer that how mitochondrial nucleoids trespass both mitochondrial membranes, is the key question. Recently, Chen et al (PMID 35313046) described in detail the role of SAMM50 in BAK/BAX pore formation, and in regulating mtDNA release. Here, we have overexpressed SAMM50 and observed that BAK clusters together with SAMM50 and that both proteins interact (Fig. 5; Fig. S8k). Super-Resolution microscopy confirmed that SAMM50, BAK and VPS35 can coexist in the same cellular structures upon mtDNA damage (Fig 7g).

In general, we believe that mtDNA damage induces local mitochondrial membrane potential changes, together with mitochondrial cristae reorganization. These processes might induce BAK/BAK oligomerization, allowing permeabilization of the mitochondrial membranes and recruitment of endosomes, followed by elimination of damaged nucleoids. We have summarized these findings in a re-drawn proposed model (Fig. S11h).

Reviewer #2 (Remarks to the Author):

In the revised manuscript, Pla-Martin and colleagues addressed the reviewers' many concerns. One issue that I still have concerns is the EtBr interpretation. Analysis of mtDNA damage by long template

amplification is tricky. The raw data is not shown and the lower amplification may be related to the one-week treatment-related mtDNA depletion. The additional treatments with H2O2 and CCCP were good additions, but as discussed, they have pleiotropic effects.

The Long Template PCR method has been performed as a complementary method to assay mtDNA damage, as suggested previously by reviewer 1. With this experiment, we pursued to demonstrate that EtBr damages mtDNA like K320E-Twinkle expression, is however faster and causes an acute response. To more clearly show this, we have now sequenced the *Cox1* gene in purified mtDNA from cells in steady state or after 3 days treatment with EtBr. The reviewer can see in the image attached, how the sample after EtBr treatment contains multiple peaks in random regions, suggesting the presence of different PCR templates, thus confirming mtDNA base changes.

Cox1

Cox1 Sanger sequencing chromatograms obtained from purified mtDNA from control and EtBr treated

In the manuscript we decided to use the Long Run quantification method. This method has been described many times (Hunter et al., *Methods*. 2010; Rothfuss et al., *Nucleic Acid Research*. 2009; Lehle et al., *Nucleic Acid Research*. 2014; Gureev et al., *Toxicology*. 2017; Karunadharmma et al., *IOVS*. 2010; Zhu et al., *BMC* 2017) and the amplification delay has been related to increased mtDNA damage. Usually, peroxide, which induces also mtDNA depletion, has been used as a positive control, as it is known to induce mtDNA damage, even though it also induces mtDNA depletion. We would like to emphasize that in case of the Long Run PCR method, every sample was calibrated using the amplification efficiency of a short mtDNA fragment as an internal “standard”, and not by comparing to samples from the control cells. We performed several calibration curves, to verify that each standard curve matched each Ct value obtained by the Long Run PCR. We also verified that the values for the standard response curve were in the exponential phase. Each value obtained by the Long Run was normalized to its own specific standard curve, so an effect of the observed depletion on the results can be excluded. Nevertheless, we provide a document containing all the raw data, where the data regarding the different standard curves and the formula for each linear regression is listed.

Reviewer #3 (Remarks to the Author):

In this revised manuscript by Pla-Martin et al., the authors have added new data and addressed some of my concerns. However, how ATAD3 and SAMM50 acts in this pathway is not described. What does these proteins really do? Is their role(s) in this pathway the same as already reported for ATAD3 and SAMM50?

In this new revised version, we are showing new key experiments that we have done to figure out the specific role of ATAD3 and SAMM50. We have analyzed mitochondrial changes in regards of cristae morphology and membrane potential upon mtDNA damage. We have found that mtDNA damage dramatically disturbs cristae morphology but only produce local changes in membrane potential (Fig 4j and k, Fig S7, S8f). Moreover, we have seen that the interaction between ATAD3 and SAMM50 is disrupted by EtBr (Fig. 4i). These data suggests that the role of these two proteins might be related to maintaining mitochondrial inner and outer contact sites and thus bridge the communication between mitochondrial matrix and cytoplasm. We do not think that the role of ATAD3 and SAMM50 in this process is similar or even the same as the ones already described. In addition, regarding the role of ATAD3 in mitophagy, importantly this function was described for ATAD3B, which is an ortholog gene only present in primates. ATAD3B is located in the mitochondrial inner membrane and translocates to the mitochondrial outer membrane in the presence of mtDNA oxidative damage (Shu et al., 2021. PMID 33665835), thus serving as a recruitment platform for autophagy components. In murine cells, only one form of ATAD3 is described. We have tried to visualize changes in murine ATAD3 topology by proteinase K experiments (see figure below). We observed an extra band below the ATAD3 protein only present in cells treated with H₂O₂ or EtBr even before Proteinase K treatment, which could indicate some type of proteolysis derived from the stress related to signalization. However, as we did not observe prominent changes, we believe that this observation is still too preliminary to be included in the manuscript.

Proteinase K treatment of isolated mitochondria from C2C12 cells treated with the indicated compounds.

With regard to SAMM50, we believe that our mechanisms might be complementary to the one described in Abudu et al., 2021. On the one side, SAMM50 is necessary to recruit the retromer VPS35, but more importantly, SAMM50 seems to regulate BAK activity, which will execute the elimination of mitochondrial nucleoids (see above). Recently, Chen et al., 2022, (PMID 35313046) showed in detail how SAMM50 regulates the BAK/BAK pore and controls mtDNA leakage. We have performed new experiments confirming some of the data observed by Chen and collaborators, for instance SAMM50 and BAK co-immunoprecipitation (Fig S8k), however to a much lower extent in our hands. It is worth to mention that the levels of SAMM50 overexpression we achieved is low, which could influence the degree of co-immunoprecipitation.

In addition, we have seen that SAMM50 and BAK distribution changes upon EtBr. Interestingly, super-resolution microscopy confirmed that VPS35, BAK and SAMM50 can be found in the same cellular compartment upon mtDNA damage. Hence, we believe that what we are observing here is a combination of the roles described already for SAMM50, on the one hand, as controlling membrane architecture and

regulating BAK function, and on the other hand, recruiting the cytosolic part of the proteins necessary for quality control.

Moreover, SAMM50 have important role in mitochondrial membrane protein insertion and cristae architecture. The effect on mtDNA clearance may be an indirect effect of SAMM50 in membrane protein integration and cristae stability. SAMM50 and ATAD3 involvement in this pathway as stated in this manuscript was determined through their interaction with Twinkle. I asked in my first review if their interaction with twinkle is relevant for their role in mtDNA clearance. You have not clarified this. This is quite simple, map out their binding to Twinkle using the co-IP experiments. If this is not possible because they do not actually bind directly, use mutants already shown to act in the already described pathways for ATAD3 (mitophagy of mtDNA) and SAMM50 (piecemeal mitophagy) for reconstitution in ATAD3 and SAMM50 KD cells to rescue mtDNA clearance. This will give a clearer understanding if this is a different pathway or those already described.

We apologize for not addressing this point before, but in our opinion, this is not an easy question. The reviewer emphasizes to decipher the role of Twinkle in mtDNA depletion. We used this protein as a target to label nucleoids and at the same time damage mtDNA with the K320E mutant. Generating Twinkle deletion constructs to map out Twinkle interaction residues will definitely impair its main function, which is mtDNA unwinding, and will thus have multiple pleiotropic effects. However, and as suggested by the reviewer, we are providing new data here using deletion constructs of ATAD3 and SAMM50, which we hope will clarify the reviewer's concerns.

In regards of ATAD3, we have generated cells expressing ATAD3-HA and ATAD3-HA (Δ IMS-ATAD3-HA) lacking the Nt part of the protein, a domain which is involved in contacting the inner side of the mitochondrial outer membrane. Even after elimination of this domain, ATAD3 and SAMM50 interaction still exists, suggesting that both proteins interact through the intermembrane domain of ATAD3 (Fig 4h), probably being part of a larger multiprotein complex in the mitochondrial membrane described recently by Chen and collaborators (Chen et al., 2022, PMID 35313046). In addition, we have reintroduced these two versions of ATAD3 in *Atad3* knock-down cells as the reviewer suggested. In agreement with our previous results, we were able to recover mtDNA depletion and relative mitochondrial nucleoid size (Fig 5d and S8j), confirming that, indeed, ATAD3 is essential to distribute and eliminate mtDNA.

In the case of SAMM50, we found the reviewer's concerns more difficult to address. SAMM50 knock-down causes mtDNA leakage, as described very recently in Chen et al., 2022, PMID 35313046, an issue that we also observed in *Samm50* KD cells (Fig 5j). Therefore, trying to recover specific mtDNA elimination by overexpressing SAMM50 in *Samm50* KD cells is not possible in the way the reviewer suggests. Hence, we cannot perform the experiment, since we cannot recover a process which is not blocked, as it happens for SAMM50 dependent piecemeal mitophagy, but rather enhanced, leading to mtDNA release. However, as suggested by the reviewer, we have generated new cells expressing SAMM50-HA and Δ POTRA-SAMM50-HA, a protein lacking the Nt and POTRA domain (aa1-150), as described in Abudu et al., 2021 (PMID 34037656). Unfortunately, we have tried to express these constructs in *Samm50* KD cells to analyze some parameters, but virus transduction leads to cell death, surprisingly only in *Samm50* KD cells. Nevertheless, we have transduced wt cells with SAMM50 constructs, and addressed the relationship between SAMM50 and BAK, as suggested by the data published in Chen et al., 2022 (PMID 35313046), but also with VPS35. We found that SAMM50 controls mitochondrial distribution of BAK and that this is not the case for Δ POTRA-SAMM50-HA (Fig 5h, i). In addition, super-resolution microscopy also showed that Δ POTRA-SAMM50-HA

failed to localize with VPS35. Hence, our data also suggests that the POTRA domain is essential, not only for BAK interaction but also to recruit VPS35 (Fig S11f and g), which is not surprising as Δ POTRA-SAMM50-HA lacks also the Nt domain. In general, we now provide new insights about mtDNA release and how SAMM50 regulates this process by serving as a bridge between mitochondria and endosomes.

In addition, the authors make too generalized statements regarding MDVs, and this needed to be modified. For example, the statement in line 524 to 525 “In addition, recently, it has been demonstrated that mtDNA cannot be localized inside of MDVs” is misleading and should be modified. I advised the authors to read the paper by König et al., 2021, PMID 34873283 properly. König et al., 2021 described the proteome of steady state TOM+ MDVs, and they stated that these MDVs may be different from stress/signal induced MDVs and other classes of MDVs. They then stated that steady state TOM+ MDVs do not contain mtDNA because the presence of mtDNA could not be validated, so is the presence of GTPases MFN1 and OPA1. We do not yet know if other classes of MDVs such as stress/signal induced MDVs contain mtDNA. The current understanding is that there are several classes of MDVs, and the proteome of stress/signal induce MDVs and other classes of MDVs including TOMM20-, PDH+ and MAPL+ MDVs have not yet been described. What you report here seem to be stress induced mtDNA removal in VPS35 positive vesicles and this may well be different from steady state TOM+ MDVs described by König et al., 2021. But to say mtDNA cannot be localized to MDV is incorrect. There is yet no proof for that. Since you only tested MAPL-induced MDVs, you should state that this pathway is independent of MAPL-dependent MDVs.

We agree that it is possible that our mechanism shares some proteins with the MDV formation complex, and we agree, that this mechanism might be a sort of stress-induced process. However, in none of our EM experiments, either Immunogold, APEX-contrasted or CLEM, we have detected MDVs. What we have seen is that Twinkle, as a part of the nucleoid, is engulfed in autophagy- like structures or close to endosome-like particles. Nevertheless, even though we do not agree with the reviewer points, we have modified the text regarding the discussion of MDVs, and left the possibility of stressed-induced MDVs open.

Minor comments

There is no description in the legend for figure S9E and S10D (change S10E to S10D). Check the other legends properly.

We apologize for the mistake. We have modified the figure legend.

Reviewer #4 (Remarks to the Author):

In this revised version, I now believe that the presented MS experiments, together with supporting WB data, are largely sound and support the draw conclusions.

Results section 1 (Figure S2):

Protein incorporation rates are comparable between K320E and WT, no enhanced turnover of mitochondrial proteins in KE mutants

Results section 3 (Figures 4, S6):

More proteins with vesicular (and OM) annotation are enriched in KE-APEX compared to Twinkle-APEX.

SAMM50 interacts with Twinkle-KE, but not with Twinkle-WT (unless adding EtBr), i.e. SAMM50 interacts with Twinkle under mtDNA damaging conditions.

Opposite is true for ATAD3, which weakly interacts with Twinkle-WT, but loses interaction under EtBr conditions.

Q: Why do these two proteins behave differently? Do you have a hypothesis for this observation?

We appreciate this reviewer's comment. Based on our results, we think that mtDNA damage induces changes in the mitochondrial membrane protein and lipid composition, allowing signal transfer through these two mitochondrial compartments. SAMM50 is a MOM transmembrane protein and ATAD3 is a transmembrane protein of the MIM. In the new data, we show that these two proteins are important for mtDNA release by allowing mitochondrial inner and outer membrane communication. We speculate that changes in the lipid/protein composition of the mitochondrial membranes might be activating a transduction signal through SAMM50 to allow BAX/BAK pore formation, recruitment of the turnover machinery and finally proper elimination of the mutated nucleoids. We discuss these mechanisms accordingly in the manuscript.

Please check whether ATAD3 is really significant in Figure S6f middle plot. I have a hard time believing that given that it is in a cloud of non-significant proteins. Please correct typo Figure S6f: q-value < -0.05

The reviewer is right. We have modified the color for ATAD3 in the figure.

Taken together and viewed from the MS angle, I believe the revised version is more coherent. However, looking at the manuscript overall, I still believe that the presented data does not fully support some conclusions. For example, I am a bit hesitant to agree that the presented data clearly shows the proposed role of ATAD3.

Southern Blot analysis in alkali conditions for different mtDNA topologies. Image was generated by Genevieve Trombly. Institute of Experimental Epileptology and Cognition Research. Bonn

Based on our observations, we believe that ATAD3 has a structural role and might be involved in signalization. We hope that with our new data, its involvement in the process is now more convincing. We draw our conclusions based on our own observations but also considering current knowledge on ATAD3. ATAD3 was found to be linked to nucleoids already in 2007 (He et al., 2007. PMID 17210950). In this publication, ATAD3 was found to bind preferentially nucleoids containing supercoiled mtDNA. In the attached image, the reviewer can see a southern blot of mtDNA topologies upon different treatments. As observed in the figure, both EtBr treated or K320E-Twinkle expressing cells decrease the proportion of supercoiled mtDNA molecules, which could explain why in these conditions, Twinkle (hence, nucleoids) binding to ATAD3 is weaker.

More recently, Gerhold et al., 2015. (PMID 26478270), described that both ATAD3 and Twinkle reside in cholesterol rich mitochondrial membrane sub-compartments, in connection to mitochondria-ER junctions. Interestingly, these areas have been shown to be a platform for the recruitment of autophagy related proteins (for a recent review, please see Yang et al., 2020. PMID 32766245). Lately, ATAD3 has been shown to control mitochondrial cristae morphology (Arguello et al., 2021. Cell Reports. PMID 34936866) and to form a complex with SAMM50, regulating mtDNA stability (Chen et al., 2022. Hepatology. PMID 35313046). In contrast to the role of ATAD3B in mtDNA-mitophagy (Shu et

al., 2021. PMID 33665835), we cannot see any changes in ATAD3 topology upon mtDNA damage (see response to reviewer 1). Furthermore, we observe changes in ATAD3 and SAMM50 interaction (Fig 4). Moreover, we have also now re-introduced ATAD3 in Atad3 KD cells and recovered both mtDNA elimination and nucleoid relative size, confirming the prominent role of ATAD3 in the process. How mtDNA damage is signaled through ATAD3 is still unknown but we will address this in future research.

REVIEWERS' COMMENTS

Reviewer #1 (Remarks to the Author):

The revised manuscript quality has been improved, and most of my concerns were addressed and resolved.

Reviewer #2 (Remarks to the Author):

Most responses are adequate. Many questions remain on the molecular mechanisms though. The remaining issue I have is that the KD of ATAD3 or SAMM50 not causing a mitochondrial morphology change is difficult to square with the previous publication on these proteins.